# Deep learning reveals endogenous sterols as allosteric modulators of the GPCR–Gα interface

Sanjay Kumar Mohanty[1†], Aayushi Mittal[1†], Namra Farooqi[2,3†], Aakash Gaur[2,4], Subhadeep Duari[1], Saveena Solanki[1], Anmol Kumar Sharma[1], Sakshi Arora[1], Suvendu Kumar[1], Vishakha Gautam[1], Nilesh Kumar Dixit[1], Karthika Subramanian[1], Tarini Shankar Ghosh[1], Debarka Sengupta[1,5], Shashi Kumar Gupta[2,4], Arul Natarajan Murugan[1], Deepak Sharma[2,3]*, Gaurav Ahuja[1,5]*

[1]Department of Computational Biology, Indraprastha Institute of Information Technology-Delhi (IIIT-Delhi), New Delhi, India; [2]Academy of Scientific and Innovative Research (AcSIR), Ghaziabad, India; [3]CSIR-Institute of Microbial Technology, Sector-39A, Chandigarh, India; [4]Pharmacology Division, CSIR-Central Drug Research Institute, Lucknow, India; [5]Infosys Centre for AI, Indraprastha Institute of Information Technology-Delhi (IIIT-Delhi), Delhi, India

*For correspondence:
deepaks@imtech.res.in (DS);
gaurav.ahuja@iiitd.ac.in (GA)

[†]These authors contributed equally to this work

Competing interest: The authors declare that no competing interests exist.

## eLife Assessment

The authors present a set of wrappers around previously developed software and machine-learning toolkits, and demonstrate their use in identifying endogenous sterols binding to a GPCR. The resulting pipeline is potentially **useful** for molecular pharmacology researchers due to its accessibility and ease of use. However, the evidence supporting the GPCR-related findings remains **incomplete**, as the machine-learning model shows indications of overfitting, and no direct ligand-binding assays are provided for validation.

**Abstract** Endogenous intracellular allosteric modulators of GPCRs remain largely unexplored, with limited binding and phenotype data available. This gap arises from the lack of robust computational methods for unbiased cavity identification, cavity-specific ligand design, synthesis, and validation across GPCR topology. Here, we developed Gcoupler, an AI-driven generalized computational toolkit that leverages an integrative approach combining de novo ligand design, statistical methods, Graph Neural Networks, and bioactivity-based ligand prioritization for rationally predicting high-affinity ligands. Using Gcoupler, we interrogated intracellular metabolites that target and regulate the GPCR–Gα interface (Ste2p–Gpa1p), affecting pheromone-induced programmed cell death in yeast. Our computational analysis, complemented by experimental validations, including genetic screening, multi-omics, site-directed mutagenesis, biochemical assays, and physiological readouts, identified endogenous hydrophobic metabolites, notably sterols, as direct intracellular allosteric modulators of Ste2p. Molecular simulations coupled with biochemical signaling assessment in site-directed Ste2p mutants further confirmed that metabolites binding to GPCR–Gα obstruct downstream signaling, possibly via a cohesive effect. Finally, by utilizing isoproterenol-induced, GPCR-mediated human and neonatal rat cardiac hypertrophy models, we observed that elevated metabolite levels attenuate hypertrophic response, reinforcing the evolutionary relevance of this mechanism.

## Introduction

G-protein-coupled receptors (GPCRs) are critical regulators of cellular processes and thus represent prime drug targets (*Yang et al., 2021*; *Alhosaini et al., 2021*; *Calebiro et al., 2021*). While traditional GPCR-targeted therapies focus on orthosteric sites, recent advances have revealed allosteric sites offering novel therapeutic avenues (*Wold and Zhou, 2018*; *Bourque et al., 2022*; *Bourque et al., 2022*; *Rees et al., 2002*; *Leach et al., 2007*). Although exogenous synthetic allosteric modulators are known, endogenous counterparts remain poorly characterized (*van der Westhuizen et al., 2015*; *Stornaiuolo et al., 2015*; *Doller, 2017*; *Reyes-Alcaraz et al., 2020*; *Zhang et al., 2015*; *O'Callaghan et al., 2012*). Developing high-affinity endogenous modulators requires integrating structure-based design, artificial intelligence (AI), and assays, yet traditional approaches like SAR analysis are hampered by limited GPCR allosteric modulator data (*Basith et al., 2018*; *Gupta et al., 2021*). While experimental techniques like FRET and BRET can validate allosteric compounds, their use in high-throughput screening for novel intracellular modulators is challenging (*Zhou et al., 2021*; *Hoffmann and Bünemann, 2010*; *Jaeger et al., 2010*). Identifying endogenous GPCR allosteric modulators is further complicated by factors like incomplete GPCR topology data (*Congreve et al., 2020*) and vast chemical space (*Topiol, 2018a*; *Yang et al., 2021*). This necessitates a hybrid computational approach combining allosteric site prediction, de novo ligand synthesis, and efficient screening, potentially enhanced by AI. Existing de novo drug design tools often lack practical applicability for this purpose due to computational limitations and high technical demands (*Spiegel and Durrant, 2020*; *Sicho et al., 2021*; *Li et al., 2021*; *Böhm, 1992*; *Nishibata and Itai, 1991*; *Pearlman and Murcko, 1993*; *Yuan et al., 2011*).

To overcome these challenges, we developed Gcoupler, a software suite (available as a Python package and Docker image) that integrates structural biology, statistical methods, and deep learning to identify GPCR allosteric modulators. We demonstrated the usability and applicability of Gcoupler in identifying novel endogenous modulators of GPCRs by exploiting the α-pheromone (α-factor)-induced mating or programmed cell death (PCD) pathway of *Saccharomyces cerevisiae*. Notably, it is well documented that the highly elevated, non-physiological levels of α-factor trigger PCD in less than half of the MATa population (*Zhang et al., 2006*). Moreover, an equivalent concentration of α-factor triggers distinct PCD kinetics across distinct laboratory strains; for example, BY4741 is more resistant than the W303 strain (*Sokolov et al., 2020*). We, therefore, hypothesized that a subset of pheromone-resistant cells might regulate the Ste2p-mediated PCD signaling via the endogenous intracellular metabolites by operating at the Ste2–Gα-binding interface. Using Gcoupler, we identified a subset of intracellular metabolites that could potentially bind to Ste2p (GPCR) at the Gpa1 (Gα)-binding interface and obstruct the downstream signaling. Our computational results further suggest that hydrophobic ligands such as sterols strengthen the Ste2p–Gpa1p binding and might trigger a cohesive response that potentially obstructs downstream signaling. Experimental evidence further supported these findings that the elevated intracellular levels of these metabolites rescue the pheromone-induced PCD. To evaluate the evolutionary conservation and possible clinically relevant translation of this mechanism, we tested these metabolites on human and rat isoproterenol-induced, GPCR-mediated cardiac hypertrophy model systems and observed attenuated response in the cardiomyocytes pretreated with GPCR–Gα-protein interface modulating metabolites.

## Results

### Development and validation of Gcoupler

Designing novel target molecules by integrating the topological, chemical, and physical attributes of protein cavities necessitates advanced neural networks. While existing approaches like Bicyclic Generative Adversarial Networks (BicycleGANs) (*Skalic et al., 2019*) and Recurrent Neural Networks (RNNs) (*Xu et al., 2021*) have demonstrated potential, end-to-end standalone tools for GPCR-specific ligand design remain scarce. To address this, we developed the Gcoupler and provided it to the community as a Python Package and a Docker image. Gcoupler adopts an integrative approach utilizing structure-based, cavity-dependent de novo ligand design, robust statistical methods, and highly powerful Graph Neural Networks. Gcoupler consists of four interconnected modules, that is, Synthesizer, Authenticator, Generator, and BioRanker, that collectively impart a smoother, user-friendly, and minimalistic experience for the end-to-end de novo ligand design.

Synthesizer, the first module of Gcoupler, takes a protein structure as input in Protein Data Bank (PDB) format and identifies putative cavities across the protein surface, providing users with the flexibility to select cavities based on druggability scores or user-supplied critical residues. Since cavity-dependent molecule generation mainly depends on the chemical composition and geometric constraints of the cavity, it is, therefore, indispensable to select the cavity for the downstream steps considering its chemical nature (hydrophobicity/hydrophilicity) and functional relevance (proximity to the active site or residue composition), among others. Accounting for these, Gcoupler offers flexibility to the users to select either of its predicted cavities based on the user-supplied critical residue or by user-supplied cavity information (amino acids) using third-party software (e.g., Pocketome) (*Hedderich et al., 2022*). To enhance user experience, Gcoupler computes and outputs all identified cavities along with their druggability scores using LigBuilder's V3 (*Yuan et al., 2020*) cavity module. Briefly, these druggability scores consider solvent accessibility, cavity exposure or burial, and detected pharmacophores and cavities, which are further prioritized based on this score. Post-cavity selection, the Synthesizer module generates cavity-specific ligands influenced by topology and pharmacophores, outputting SMILES, cavity coordinates, and other requisite files to downstream modules for further steps (*Figure 1a*). The chemical composition of the in silico synthesized ligands by the Synthesizer module is influenced by the cavity topology (3D) and its composition (pharmacophores). Noteworthy, the Synthesizer module of Gcoupler employs LigBuilder V3 (*Yuan et al., 2020*), which utilizes the genetic algorithm for the in silico ligand synthesis. Notably, the fragment library of LigBuilder, comprising 177 distinct molecular fragments in Mol2 format, allows the selection of multiple seed structures and extensions that best complement the cavity pharmacophores throughout multiple iterative runs. For each run, once a seed structure is confirmed, Gcoupler employs a hybrid approach of the Growing and Linking modes of the LigBuilder build module, enabling the stepwise addition of small fragments to the seed structure within the binding pocket of the target GPCR to build synthetic ligands. Gcoupler generates 500 unique molecules by default, though it can also be user-defined. The Synthesizer module of Gcoupler enhances LigBuilder V3 practical applicability through automation, dynamic adaptability, and abstraction. This allows for more efficient and targeted ligand generation, even in challenging design scenarios for GPCR ligand design. However, it lacks user-defined library screening, proposes synthetically challenging molecules, and often requires post-processing to isolate High-Affinity Binders () from a broad affinity range of synthetically designed compounds.

To address this limitation, a second module was added to Gcoupler, termed Authenticator. This module processes output files from the Synthesizer module, conducting downstream validation steps and preparing results for constructing deep learning-based classification models (third module). The Authenticator requires input protein 3D structure in PDB format, cavity coordinates, and all silico-generated molecules from the Synthesizer module. Authenticator utilizes this information to further segregate the synthesized compounds into HABs and Low-Affinity Binders (LABs) by leveraging a structure-based virtual screening approach (AutoDock Vina) (*Trott and Olson, 2010*) and statistically backed hypothesis testing for distribution comparisons (*Figure 1a*). The Authenticator module outputs the free binding energies of all the generated compounds, which further segregates the compounds into HABs and LABs by the statistical submodule while ensuring the optimal binding energy threshold and class balance. Of note, the Authenticator is also capable of leveraging the Empirical Cumulative Distribution Function (ECDF) for binding energy distribution comparisons of HABs and LABs and performs the Kolmogorov–Smirnov test (*Berger and Zhou, 2014*), Epps–Singleton test (*Goerg and Kaiser, 2009*), and Anderson–Darling test (*Engmann and Cousineau, 2011*) for hypothesis testing. This expanded array of statistical tests allows users to employ methodologies that best suit their data distribution characteristics, ensuring robust and comprehensive analyses. Moreover, the Authenticator module incorporates a unique feature for decoy synthesis using HABs. This functionality enables the generation of a negative dataset in scenarios where the Synthesizer module fails to produce an optimal number of LABs. By synthesizing decoys from HABs, users can effectively balance their datasets, enhancing the reliability of downstream analyses. Lastly, the Authenticator module also accommodates user-supplied negative datasets as an alternative to LABs (*Mysinger et al., 2012*). This feature provides users with the flexibility to incorporate external data sources, enabling robust prediction model building by the subsequent Generator module.

The Generator, the third module, employs state-of-the-art GNN models such as Graph Convolution Model (GCM), Graph Convolution Network (GCN), Attentive FP (AFP), and Graph Attention

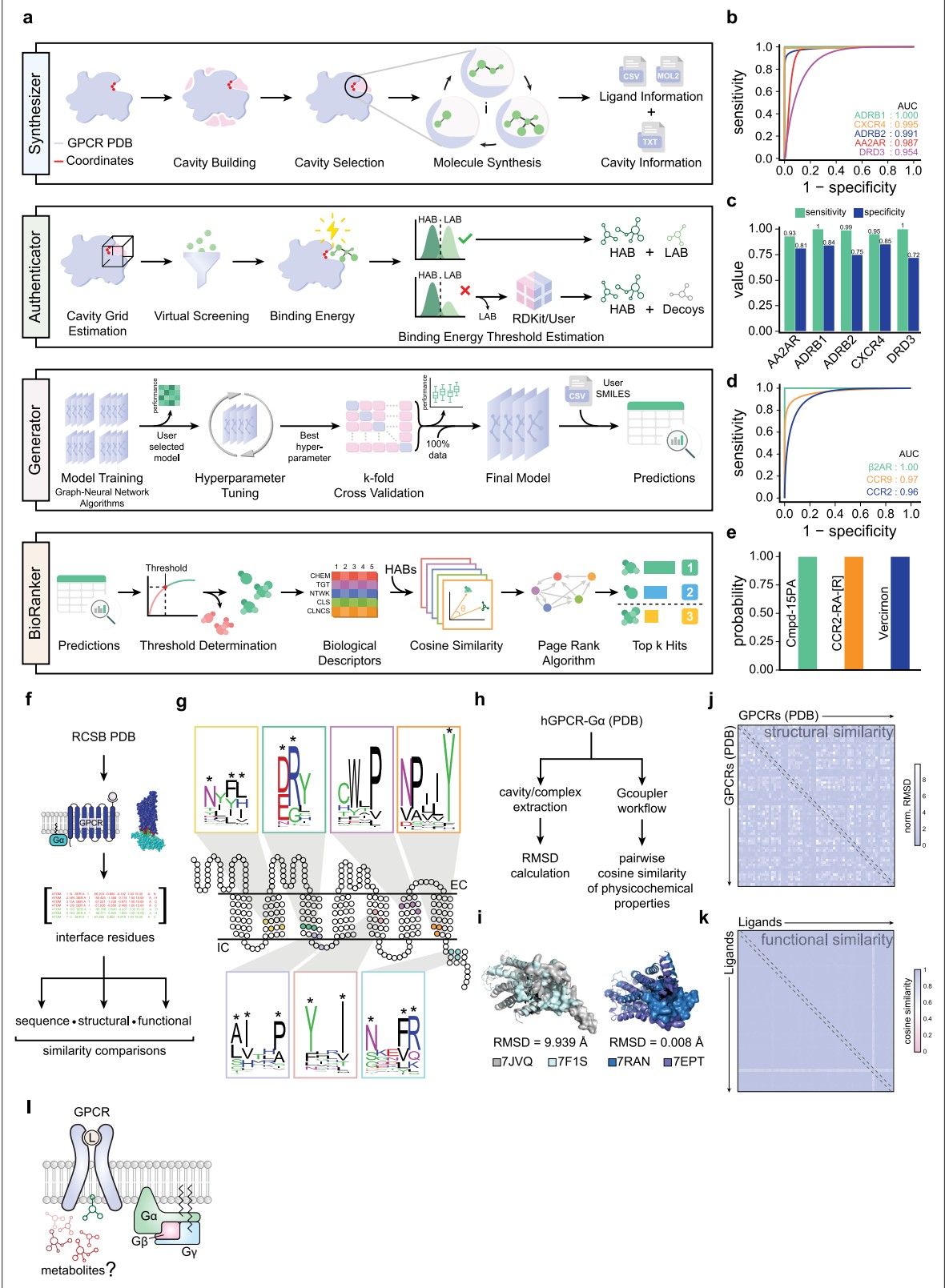

**Figure 1.** Development, benchmarking, and validation of Gcoupler computational framework. (**a**) Schematic workflow depicting different modules of the Gcoupler package. Of note, Gcoupler possesses four major modules, that is, Synthesizer, Authenticator, Generator, and BioRanker. (**b**) AUC–ROC curves of the finally selected model for each of the indicated GPCRs. Note: Experimentally validated active ligands and decoys were used in the testing dataset. (**c**) Bar graphs depicting the sensitivities and specificities of the indicated GPCRs with experimentally validated active ligands and reported

*Figure 1 continued*

decoys. (**d**) The AUC–ROC curve indicates the model's performance in the indicated conditions. (**e**) Bar graphs indicating the prediction probabilities for each experimentally validated ligand. (**f**) Schematic workflow illustrates the steps in measuring and comparing the structural conservation of the GPCR–Gα-protein interfaces across human GPCRs. (**g**) Snake plot depicting the standard human GPCR two-dimensional sequence level information. Conserved motifs of the GPCR–Gα-protein interfaces are depicted as WebLogo. Asterisks represent residues of conserved motifs present in the GPCRs–Gα-protein interfaces. Of note, the location of the motifs indicated in the exemplary GPCR snake plot is approximated. (**h**) Schematic workflow illustrates the steps in measuring and comparing the structural conservation of the GPCR–Gα-protein interfaces across human GPCRs. (**i**) Representative structures of the proteins depicting highly conserved (low root mean square deviation [RMSD]) and highly divergent (high RMSD) GPCR–Gα-protein interfaces. PDB accession numbers are indicated at the bottom. (**j**) Heatmap depicting the RMSD values obtained by comparing all the GPCR–Gα-protein interfaces of the available human GPCRs from the protein databank. Of note, the RMSD of the Gα–protein cavity was normalized with the RMSDs of the respective whole proteins across all pairwise comparisons. (**k**) Heatmap depicting the pairwise cosine similarities between the in silico synthesized ligands of the GPCR–Gα-protein interfaces of the available human GPCRs using Gcoupler. (**l**) Schematic diagram depicting the hypothesis that the intracellular metabolites could allosterically modulate the GPCR–Gα interaction.

The online version of this article includes the following figure supplement(s) for figure 1:

**Figure supplement 1.** Gcoupler benchmarking using experimentally validated orthosteric ligands of GPCRs.

**Figure supplement 2.** Gcoupler benchmarking using experimentally validated allosteric ligands of GPCRs.

**Figure supplement 3.** Run time comparison of Gcoupler and AutoDock.

Network (GAT) to construct predictive classifiers using Authenticator-informed classes. These GNN algorithms are tailored to extract features from the graph structure of the compounds generated by the Synthesizer and apply them to the classification task by leveraging Authenticator-informed class information. For instance, the GCM assimilates features by analyzing neighboring nodes, while the GCN extracts features through a convolutional process. The AFP model focuses attention on specific graph segments, and the GAT employs attention mechanisms to learn node representations. By default, Generator tests all four models using standard hyperparameters provided by the DeepChem framework (https://deepchem.io/), offering a baseline performance comparison across architectures. This includes pre-defined choices for node features, edge attributes, message-passing layers, pooling strategies, activation functions, and dropout values, ensuring reproducibility and consistency. All models are trained with binary cross-entropy loss and support default settings for early stopping, learning rate, and batch standardization where applicable. Gcoupler provides off-the-shelf hyperparameter tuning to ensure adequate training, which is essential for optimizing model performance. After selecting the best parameters and classification algorithm, Gcoupler further ensures the mitigation of overfitting and provides a more precise estimate of model performance through *k*-fold cross-validation. Notably, by default, Gcoupler employs threefold cross-validation, but users can adjust this parameter.

Finally, BioRanker, the last module, prioritizes ligands through statistical and bioactivity-based tools. The first level ranking offered by BioRanker is composed of a statistical tool that encompasses two distinct algorithms, namely *G*-means and Youden's *J* statistics, to assist users in identifying the optimal probability threshold, thereby refining the selection of high-confidence hit compounds (*Figure 1—figure supplement 1a*). Additionally, bioactivity embeddings computed via Signaturizer (*Bertoni et al., 2021*) enable multi-activity-based ranking using a modified PageRank algorithm. Briefly, the bioactivity descriptors of the predicted compounds are projected onto various biological activity spaces, including Chemistry, Targets, Networks, Cells, and Clinics, by performing pairwise cosine similarity comparisons with HABs. The PageRank algorithm is then applied for activity-specific ranking and supports multi-activity-based ranking for sequential screening based on user-defined biological properties. BioRanker also offers flexibility through customizable probability thresholds, enabling stringent or relaxed selection of compounds. Users can also input SMILES representations for direct screening, bypassing prediction probabilities. Taken together, Gcoupler is a versatile platform supporting user-defined inputs, third-party tools for cavity selection, and customizable statistical analyses, enhancing its adaptability for diverse ligand design and screening tasks. This integrated framework streamlines cavity-specific ligand design, screening, and ranking, providing a comprehensive solution for GPCR-targeted drug discovery.

To evaluate Gcoupler's performance, we tested its modules across five GPCRs (AA2AR, ADRB1, ADRB2, CXCR4, and DRD3) using experimentally validated ligands and matched decoys from the DUD-E dataset (*Mysinger et al., 2012*). The DUD-E datasets contain five GPCRs alongside information

about their cavity coordinates, positive ligands, and decoys (https://dude.docking.org/subsets/gpcr). We used these five GPCRs as independent samples to evaluate different modules and sub-modules of Gcoupler. We first checked whether the cavity search algorithm of Synthesizer could accurately detect a given orthosteric ligand-binding site for a GPCR. Gcoupler accurately identified orthosteric ligand-binding sites and additional allosteric cavities across all targets, validating its de novo cavity detection algorithm (*Figure 1—figure supplement 1b*). We next asked whether Gcoupler could also synthesize molecules similar to the reported ligands for respective orthosteric sites based on the cavity's physical, chemical, and geometric properties. For orthosteric sites, the Synthesizer module generated ~500 compounds per GPCR. Subsequently, as per the Gcoupler default workflow, the Authenticator module conducted a virtual screening of these newly synthesized compounds, segregating them into HABs and LABs. Although the Authenticator module provides flexibility in selecting an optimal threshold to distinguish HAB and LAB, we chose the default cutoff of –7 kcal/mol for AA2AR, CXCR4, and DRD3. For ADRB1 and ADRB2, we selected a threshold of –8 kcal/mol to minimize overlap in distributions and thus avoid class imbalance, a critical parameter that could influence the downstream model generation using the Generator module (*Figure 1—figure supplement 1c*). Statistical validation confirmed significant separation between these groups (p < 0.0001), enabling the Generator module to construct graph-based classification models with high values of AUC–ROC (>0.95), sensitivity, and specificity (*Figure 1b, c*, *Figure 1—figure supplement 1d, e*). These models reliably distinguished ligands from decoys, demonstrating Gcoupler's accuracy in identifying high-affinity ligands.

In addition to evaluating Gcoupler's performance for the orthosteric sites of GPCRs, we also validated its capability to identify allosteric sites and their corresponding ligands. In this case, we first gathered information about the experimentally validated GPCR–ligand complexes sourced from the PDB database. We chose three GPCR–ligand complexes (β2AR-Cmpd-15PA, CCR2-CCR2-RA-[R], and CCR9-Vercirnon) from the PDB (*Shen et al., 2023*). We removed the ligands from the PDB files and executed the standard Gcoupler workflow with default parameters. Gcoupler successfully identified allosteric binding sites and generated classification models for synthetic compounds with consistently high AUC–ROC values (>0.95) (*Figure 1d*, *Figure 1—figure supplement 2a–c*). This high level of accuracy indicates the robustness of Gcoupler's algorithms in distinguishing between true positives (allosteric ligands) and true negatives (non-binders). Projection of experimentally validated ligands onto these models further confirmed their predictive accuracy (*Figure 1e*), underscoring Gcoupler's robustness and versatility for orthosteric and allosteric ligand discovery.

Next, to evaluate the efficiency of Gcoupler, we compared its run time with the biophysics-based gold standard molecular docking (AutoDock) (*Morris et al., 2009*). To address the runtime efficiency, we first utilized the ChEMBL31 database (*Gaulton et al., 2012*) to identify GPCRs with the highest number of reported experimentally validated agonists. We selected the alpha-1A adrenergic receptor (ADRA1A) since it qualifies for this criterion and contains 993 agonists (*Figure 1—figure supplement 3a, b*). Methodologically, we followed the conventional steps of AutoDock Tools for molecular docking while keeping track of execution time for each step throughout the entire process until completion (*Figure 1—figure supplement 3a, c*). In parallel, we applied the same timestamp procedure for Gcoupler, including its individual module sub-functions (*Figure 1—figure supplement 3a–d*). Gcoupler was 13.5 times faster, leveraging its deep learning-based Generator module and AutoDock Vina's efficiency. Both methods provided comparable predictions for active compounds, demonstrating Gcoupler's speed and accuracy, making it ideal for large-scale ligand design and drug discovery (*Figure 1—figure supplement 3e–h*).

Finally, we used Gcoupler to evaluate the ligand space conservation (functional conservation) of the GPCR–Gα interface. Specifically, we aimed to explore the possibility of direct small molecule binding to the GPCR–Gα interface to modulate downstream signaling pathways. We analyzed multiple human GPCR–Gα complexes from the PDB (*Figure 1f*, *Supplementary file 1*), identified conserved motifs (DRY, CWxL, and NPxxY) and binding pockets through sequence and structural analyses (*Figure 1g*). To determine the topological similarity of the GPCR–Gα-protein interface, we undertook a detailed structural analysis across a wide array of GPCR–Gα-protein complexes. This analysis involved identifying and extracting the cavities present within each complex. By focusing on these critical regions, we aimed to assess the degree of structural conservation and quantify it through normalized root mean square deviation (RMSD) values. Specifically, the normalized RMSD values, which provide a

measure of the average distance between atoms of superimposed proteins, indicated a high degree of similarity. The mean RMSD value was found to be 1.47 Å, while the median RMSD value was even lower at 0.86 Å. These values suggest that the overall topology of the GPCR–Gα interface is well conserved across different complexes, highlighting the robustness of this interaction site (*Figure 1h–j*, *Figure 1—figure supplement 3i–k*, *Supplementary file 2*). Finally, to test whether this topological and sequence conservation also impacts the ligand profiles that could potentially bind to this interface, we performed the Gcoupler workflow on all 66 GPCRs and synthesized ~50 unique ligands per GPCR (*Figure 1h*). We next computed and compared the physicochemical properties (calculated using Mordred; *Moriwaki et al., 2018*) of these synthesized ligands and observed high cosine similarity, which further supports the functional conservation of the GPCR–Gα interface (*Figure 1h, k*, *Figure 1—figure supplement 3l*, *Supplementary file 3*). In summary, we used Gcoupler to systematically evaluate and analyze the ligand profiles of the GPCR–Gα-protein interface and observed a higher degree of sequence, topological, and functional conservation.

## Gcoupler reveals endogenous, intracellular Ste2p allosteric modulators

We next utilized Gcoupler to test the hypothesis that the intracellular metabolites could potentially and directly regulate the GPCR signaling by directly interacting with the GPCR–Gα-protein interaction interface (*Figure 1l*). To test this hypothesis, we utilized a well-characterized yeast mating pathway mediated via the Ste2p–Gpa1p interface (*Figure 2—figure supplement 1a*). We used Gcoupler to screen for such metabolites against the Yeast Metabolome Database (YMDB) (*Jewison et al., 2012*). We utilized the recently elucidated cryo-EM structure of the Ste2 protein (*Velazhahan et al., 2021*) and performed a small-scale molecular dynamics (MD) simulation by using a yeast phospholipid composition-based lipid bilayer environment (*Kaneko et al., 1976*), and subjected the simulated stable structure to the Gcoupler workflow (*Figure 2—figure supplement 1b–i*). This led to the identification of 17 potential surface cavities on Ste2p. Careful interrogation of the structurally supported Ste2p–Gpa1p interface revealed two distinct predicted cavities (annotated as IC4 (intracellular cavity 4) and IC5 (intracellular cavity 5)), collectively capturing >95% of the interface regions (*Figure 2a, b*, *Figure 2—figure supplement 1j–n*). Conservation analysis of these cavities across 14 yeast species further confirmed their structural significance (*Figure 2—figure supplement 1o*). We next synthesized ~500 in silico synthetic compounds for both IC4 and IC5, each by leveraging the Synthesizer module of Gcoupler. To test whether the chemical space of these in silico synthesized ligands is cavity-specific, we performed a stringent evaluation by comparing the chemical heterogeneity of the 100 randomly selected synthesized ligands, each from the pool of 500 for IC4 and IC5, with 100 de novo synthesized ligands for an extracellular cavity (EC1). Of note, EC1 does not harbor any overlapping residue with IC4 or IC5 and possesses distinct pharmacophore properties (*Figure 2—figure supplement 2a, d, e*). Next, we computed the atom pair fingerprints and visualized the chemical heterogeneity in the low-dimensional space using 2D and 3D PCA (*Figure 2—figure supplement 2b, c*). These results suggest that the Synthesizer module of Gcoupler generated cavity-specific ligands by leveraging both the cavity topology (3D) and its composition (pharmacophore). We further assessed the reproducibility of the Gcoupler by synthesizing 100 in silico compounds per run across five runs for the Ste2p IC4 cavity. Visualization of the chemical heterogeneity between the compounds generated via different runs in the low-dimensional space using 2D/3D PCA and pairwise Tanimoto Similarity using atom pair fingerprints suggests heterogeneous and overlapping chemical composition among the synthesized ligands across all five runs (*Figure 2—figure supplement 2f–h*).

Post these performance/reproducibility checks of Gcoupler, we segregated the 500 in silico synthesized ligands for IC4 and IC5, each into HABs and LABs by the Authenticator module of Gcoupler. Notably, the estimated binding energy threshold was set at –7 kcal/mol, a widely accepted cutoff in virtual screening (*Wong et al., 2022*; *Alanzi et al., 2024*; *Figure 2c, d*). A deep investigation into these classified synthetic compounds showed a comparable similarity between the HABs and LABs of the aforementioned target cavities, respectively (*Figure 2—figure supplement 3a–c*). The Generator module built classification models by implementing four distinct Graph Neural Network algorithms. Comparing the model performance metrics suggests that AFP outperformed other algorithms for both cavities (*Figure 2e*, *Figure 2—figure supplement 3d*). Next, we screened yeast metabolites, compiled from YMDB, on the best-performing model (hyperparameter-tuned AFP model) and predicted metabolites that could potentially bind to the IC4 and IC5 of the Ste2p–Gpa1p interface

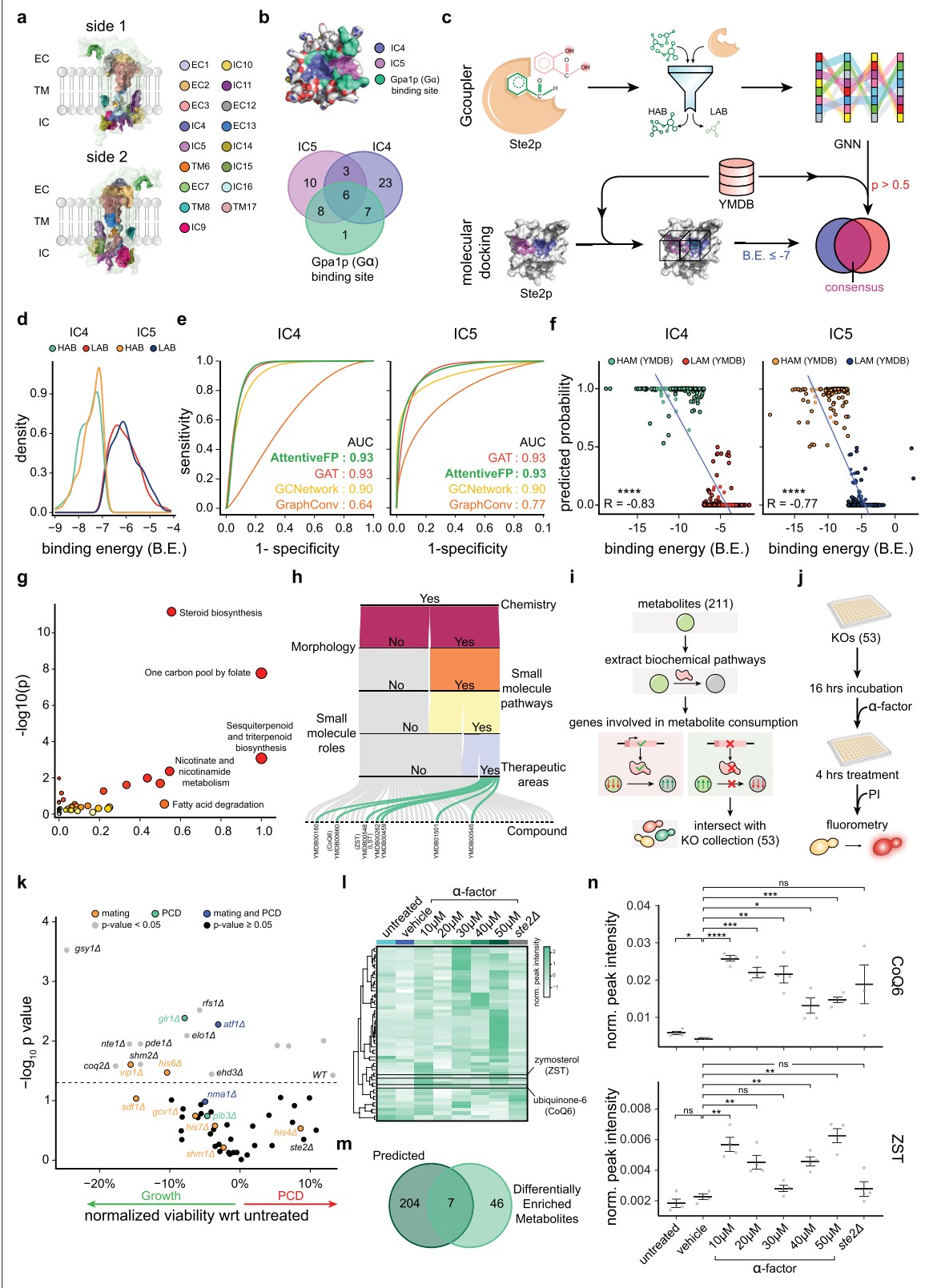

**Figure 2.** Identification of endogenous, intracellular allosteric modulators of Ste2p using Gcoupler. (**a**) Schematic diagram depicting the topology of all the cavities identified using the Synthesizer module of the Gcoupler Python package. Of note, the cavity nomenclature includes the cavity location, that is, EC (extracellular), IC (intracellular), and TM (transmembrane), succeeded by a numerical number. (**b**) Diagram depicting the three-dimensional view of the Ste2 protein, with highlighted Gα-protein-binding site (Gpa1) and the Gcoupler intracellular cavities (IC4 and IC5). The Venn diagram at the bottom

*Figure 2 continued on next page*

*Figure 2 continued*

depicts the percentage overlap at the amino acid levels between the Gα-binding site and predicted IC4 and IC5. (**c**) Schematic representation of the overall workflow used to predict the endogenous intracellular allosteric modulators of Ste2 receptor using Gcoupler and molecular docking technique. Of note, Yeast Metabolome Database (YMDB) metabolites were used as query compounds. (**d**) Overlapping density plots depicting and comparing the distributions of synthetic compounds predicted to target the IC4 and IC5 of the Ste2 receptor using the Gcoupler package. Of note, the Authenticator module of Gcoupler segregated the synthesized compound for each cavity (IC4 or IC5) into High-Affinity Binders (HABs) and Low-Affinity Binders (LABs). (**e**) AUC (area under the curve) plots representing the performance of the indicated models. Notably, the models were trained using the cavity-specific synthetic compounds generated using the Gcoupler package. (**f**) Scatter plots depicting the relationship (correlation) between the binding prediction probabilities using Gcoupler and binding free energies computed using molecular docking (AutoDock). (**g**) Scatterplot depicting the Pathway Over Representation Analysis (ORA) results of the endogenous metabolites that were predicted to bind to the GPCR–Gα-protein (Ste2p–Gpa1p) interface using both Gcoupler and molecular docking. (**h**) Alluvial plot showing five-level sub-activity spaces screening of the selected metabolites for IC4. (**i**) Schematic diagram depicting the workflow opted to narrow down on the single metabolic gene mutants. (**j**) Schematic diagram depicting the experimental design used to screen single metabolic gene mutants for α-factor-induced programmed cell death (PCD). Cell viability was assessed using a propidium iodide-based fluorometric assay. (**k**) Scatter plot depicting the impact of α-factor stimuli on cellular viability, assessed using propidium iodide-based fluorometric assay. The *y*-axis represents $-\log_{10}$(p-value) of the one-sample Student's *t*-test between the normalized PI fluorescence of untreated and treated conditions. The *x*-axis represents the percentage inhibition or increase in cellular viability, estimated using a propidium iodide-based assay. The mutants reported to be involved in mating, PCD, or both are indicated in orange, green, and blue, respectively. The statistically non-significant mutants are indicated below the dashed line in black. (**l**) Heatmap depicting the relative enrichment/de-enrichment of differentially enriched metabolites (DEMs) in the indicated conditions. Of note, four biological replicates per condition were used in the untargeted metabolomics. (**m**) Venn diagram depicting the overlap between the predicted endogenous intracellular allosteric modulators of Ste2p and DEMs identified using untargeted metabolomics. (**n**) Mean-whisker plot depicting the relative abundance of ubiquinone 6 (CoQ6) and zymosterol (ZST) in the indicated conditions. Student's *t*-test was used to compute statistical significance. Asterisks indicate statistical significance, whereas ns represents non-significance.

The online version of this article includes the following figure supplement(s) for figure 2:

**Figure supplement 1.** Ste2 protein cavities prediction using Gcoupler.

**Figure supplement 2.** Comparison of the chemical composition of cavity-specific ligands of Ste2 protein.

**Figure supplement 3.** Sanity check of Gcoupler predictions.

**Figure supplement 4.** Untargeted metabolomics and genetic screening suggest an interlink between metabolites and Ste2p-mediated programmed cell death (PCD).

---

with a binding probability cutoff of >0.5 (*Figure 2c*). To further optimize the lead metabolite list, we parallelly performed the standard molecular docking using AutoDock with YMDB metabolites against IC4 and IC5 of Ste2p, respectively, and ultimately selected the consensus metabolites (binding energy ≤–7 kcal/mol and binding probability >0.5) for the downstream analysis (*Figure 2c*, *Figure 2—figure supplement 3e*, *Supplementary file 4*). Of note, the consensus metabolites list was further segregated into HAMs (High-Affinity Metabolites) and LAMs (Low-Affinity Metabolites) based on the binding prediction probability (cutoff = 0.5). Comparative analysis of the binding prediction probabilities of Gcoupler and binding energies from Autodock of HAM and LAM revealed a significant negative correlation that further validates the authenticity of our novel approach (*Figure 2f*). Of note, as expected, HAMs and LAMs possess distinct atomic fingerprints, as indicated by the Principal Component Analysis (*Figure 2—figure supplement 3f*). Furthermore, HAMs and LAMs displayed distinct atomic fingerprints, with enriched functional groups, including R2NH, R3N, ROPO3, ROH, and ROR, observed in HAMs (*Figure 2—figure supplement 3g, h*). To gain the pathway-level information about these putative endogenous intracellular allosteric modulators of Ste2p, we performed pathway-level over-representation analysis (*Chong et al., 2018*) and observed the selective enrichment of metabolites involved in the steroid, sesquiterpenoid, and triterpenoid biosynthesis and one-carbon pool by folate pathways (*Figure 2g*). BioRanker module further pinpointed sterols, including zymosterol (ZST), ubiquinone 6 (CoQ6), and lanosterol (LST), as top candidates, exhibiting high prediction probabilities (>0.99) and structural similarity to HABs (*Figure 2h*, *Figure 2—figure supplement 3i–j*). To validate Gcoupler-identified allosteric modulators, we performed control analyses of the Authenticator and Generator modules along with blind docking of YMDB metabolites with Ste2p. In the former case, we removed the class information (HAB and LAB labels) from the in silico compounds synthesized for the IC4 cavity of the Ste2p, which resulted in a heterogeneous pool of chemical compounds. We next randomly split the training and testing data (five iterations) and build independent models. Our results suggest that compared to the Authenticator-guided data splitting (HAB and LAB), the random splitting resulted in poor model performances (*Figure 2—figure supplement 3k*), suggesting the robustness of the Authenticator module. Additionally, we also evaluated the impact of the size of

the training data on the Generator model performance. To achieve this, we randomly selected 25%, 50%, 75%, and 100% of the in silico synthesized compounds of the Ste2p IC4 cavity and built models using default parameters. Our results revealed a significant increase in the model performance with increased training data size (*Figure 2—figure supplement 3l, m*). For the latter case, we performed blind docking by AutoDock for Ste2p with the YMDB metabolites and compared these results with the cavity-specific docking via AutoDock and Gcoupler predictions. As expected, in contrast to the cavity-specific Gcoupler and AutoDock, where we observed significant segregation of the HAM and LAM at –7 kcal/mol binding energy (BE) cutoff and 0.5 as the probability cutoff of Gcoupler, we failed to observe any striking differences for the HAM in the case of blind docking (*Figure 2—figure supplement 3n*). All these rigorous control analyses and blind docking validated the reliability of Gcoupler's predictions, confirming its robustness in identifying cavity-specific modulators.

To experimentally validate the role of Gcoupler-predicted metabolites in Ste2p signaling, we performed a genetic screen of metabolic mutants. We first mapped the predicted allosteric-modulating metabolites to biochemical pathway databases (KEGG and MetaCyc) and identified the enzymes responsible for processing these metabolites (*Figure 2i*, *Figure 2—figure supplement 4a*, *Supplementary file 5*). Among the 53 single metabolic mutants (+*ste2Δ*) screened, only Ste2 was previously reported in KEGG pathways for altered mating response (*Figure 2—figure supplement 4b*). Next, we performed large-scale activity screening using α-factor-induced PCD assays with propidium iodide (PI), assuming that metabolic gene deletions would lead to intracellular accumulation of target metabolites. Briefly, selected single metabolic mutants (MATa) were grown under optimal conditions to late log phase (16 hr). Growth profile analysis revealed varied responses relative to wild-type, with most mutants exhibiting delayed growth kinetics (*Figure 2—figure supplement 4c*). Late log phase cells were subsequently treated with α-factor, and PCD induction was quantified using PI-based cell viability assays (*Figure 2j*). Notably, wild-type BY4741 strains showed significant PI fluorescence increase, indicating pheromone-induced PCD, and *STE2* knockout mutants (*ste2Δ*) showed no significant death as expected. Interestingly, most metabolic mutants (94.4%) resisted α-factor-induced cell death, with some displaying accelerated growth in the presence of α-factor, indicating crosstalk between central metabolism and Ste2 signaling (*Figure 2k*). These results indicate that a significant proportion of Gcoupler predicted metabolites could directly or indirectly influence the Ste2 signaling pathway, establishing a link between metabolism and Ste2 signaling.

To further investigate the metabolic pathways associated with PCD resistance, we performed high-resolution metabolomics on cells surviving α-factor treatment at varying concentrations (*Figure 2—figure supplement 4d*). Unbiased metabolome analysis revealed differentially enriched metabolites in the surviving population (*Figure 2l*, *Figure 2—figure supplement 4e–i*, *Supplementary file 6*). Cross-comparison analysis identified seven metabolites overlapping between Gcoupler predictions and survivor-enriched metabolites, with ubiquinone 6 (CoQ6) and zymosterol (ZST) showing prominent enrichment across all tested concentrations (*Figure 2m, n*, *Figure 2—figure supplement 4j*). Over-Representation Analysis of the differentially enriched metabolites suggests their involvement in glyoxylate and dicarboxylate metabolism, purine metabolism, and vitamin B6 metabolism, among others. (*Figure 2—figure supplement 4k*). Taken together, the findings from genetic screening and untargeted metabolomics hint toward the interplay between the central metabolism and Ste2 signaling, with computationally predicted metabolites like ZST and CoQ6 potentially conferring resistance to α-factor-induced PCD.

## Elevated endogenous metabolic levels selectively inhibit GPCR signaling

To evaluate the stability of the interactions of zymosterol (ZST), lanosterol (LST), and ubiquinone 6 (CoQ6) at the Ste2p–Gpa1p interface, we performed three independent replicates of MD simulations (short run of 100 ns) of the Ste2p–metabolite complex for both cavities (*Figure 3a–c*, *Figure 3—figure supplement 1a*). MD simulation results suggest that the interactions between the metabolites and Ste2p at the Ste2p–Gpa1p interface are thermodynamically stable in almost all cases across both the cavities (IC4 and IC5), except for the ubiquinone 6 (CoQ6), which harbored a fluctuating RMSD over the simulation timeframe (*Figure 3c*). Notably, fluctuating RMSD is observed only in the case of IC5, while it is within the permissive range for IC4 (*Figure 3c*, *Figure 3—figure supplement 1b*). We further evaluated IC4 by performing a longer simulation run for 550 ns with these three Ste2p–metabolite

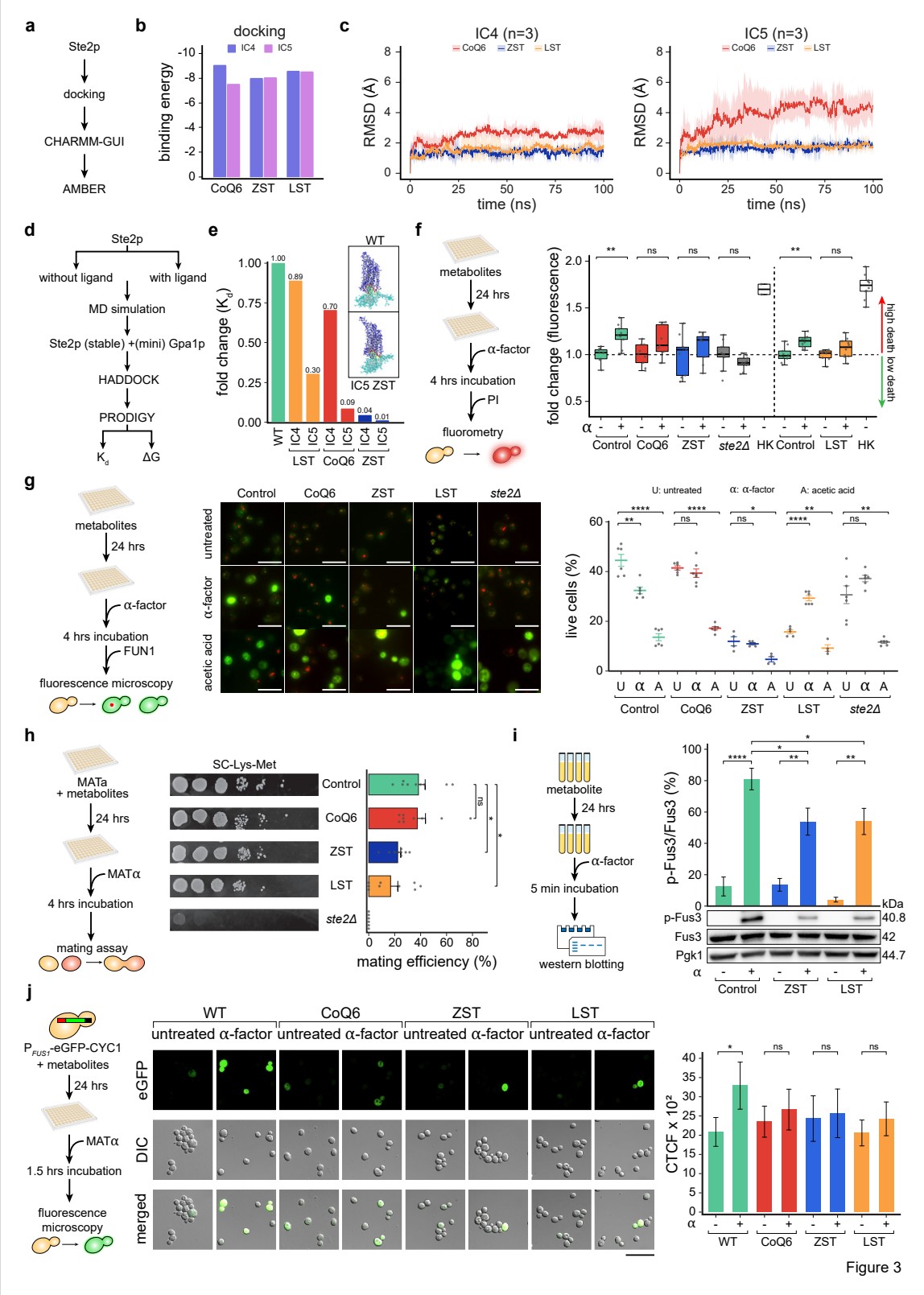

**Figure 3.** Elevated endogenous metabolite levels stabilize Ste2p–Gpa1p interactions and selectively inhibit GPCR signaling. (**a**) Scheme representing the key steps opted for preparing Ste2p structure for downstream computational analysis. (**b**) Bar plots depicting the binding energies obtained by the docking of Ste2p and indicated metabolites across IC4 and IC5. (**c**) Line plots depicting the root mean square deviation (RMSD) changes over simulation timeframes from the three independent replicates of the indicated conditions in the indicated conditions. The spread of the data is indicated

*Figure 3 continued on next page*

*Figure 3 continued*

as standard deviation (SD). Notably, RMSD is provided in Angstroms (Å), whereas the simulation time is in nanoseconds (ns). (**d**) Workflow depicting the steps involved in Ste2p-miniG-protein docking using HADDOCK and PRODIGY web servers. (**e**) Bar plots depicting the fold change of the dissociation constant ($K_d$) in the indicated conditions. Notably, fold change was computed with respect to the wild-type condition (Ste2p-miniG-protein). Inlets represent molecular representations of Ste2p-miniG-protein and the highlighted interface residues. (**f**) The schematic diagram depicts the experimental workflow used to quantify α-factor-induced programmed cell death (PCD) using a propidium iodide-based cell viability fluorometric assay. Box plot on the right depicting the rescue from the α-factor-induced PCD in the indicated conditions as inferred using propidium iodide-based cell viability fluorometric assay (*n* = 9 or 10 biological replicates; heat-killed = 2). The *y*-axis represents the fold change of the propidium iodide fluorescence values with respect to their respective controls. Mann–Whitney *U* test was used to calculate statistical significance. Asterisks indicate statistical significance, whereas ns represents non-significance. (**g**) Schematic representation (left) of the experimental approach used to measure cell vitality and viability using microscopy-based FUN1 staining. Representative micrographs (right) depicting the FUN1 staining results in the indicated conditions, Scale 10 μm. Mean-whisker plot depicting the relative proportion of the vital and viable yeast cells observed using FUN1 staining in the indicated conditions (*n* = 3 biological replicates). A Student's *t*-test was used to compute statistical significance. Asterisks indicate statistical significance, whereas ns represents non-significance. Error bars represent the standard error of the mean (SEM). (**h**) Schematic representation (left) of the experimental design for the mating assay (*n* = 3 biological replicates, each with three technical replicates). MATa yeast cells were pre-loaded with the metabolites and then mated with MATα cells to evaluate the mating efficiency. Representative micrographs in the middle qualitatively depict the mating efficiency in the indicated conditions. The bar plots on the right depict the mating efficiency (mean ± SEM) in the indicated conditions. Student's *t*-test was used to compute statistical significance. Asterisks indicate statistical significance, whereas ns represents non-significance. (**i**) Schematic representation depicting the experimental design of phospho-MAPK activity-based Western blot. Bar plots depicting the p-Fus3 levels (mean ± SEM; n = 5–6 biological replicates after IQR-based outlier removal) in the indicated conditions. Error bars represent the standard error of the mean (SEM). A Student's *t*-test was used to compute statistical significance. Asterisks indicate statistical significance, whereas ns represents non-significance. (**j**) Schematic representation (left) of the experimental approach used to measure the fluorescence in P$_{FUS1}$-eGFP-CYC1 yeast cells. Representative micrographs (right) depicting the eGFP expression in the yeast cells in the indicated conditions. Scale 20 μm. Bar plot depicting the Corrected Total Cell Fluorescence (CTCF) value (mean ± SEM; n = 3 biological replicates) in the indicated conditions. A Student's *t*-test was used to compute statistical significance. Asterisks indicate statistical significance, whereas ns represents non-significance.

The online version of this article includes the following source data and figure supplement(s) for figure 3:

**Source data 1.** PDF file containing original western blots of p-Fus3, Fus3, and Pgk1 displayed in *Figure 3i*.

**Source data 2.** Original files for western blot analysis of p-Fus3, Fus3, and Pgk1 displayed in *Figure 3i*.

**Figure supplement 1.** Docking and molecular dynamics (MD) simulations suggest the stability of the Ste2–metabolite complex.

**Figure supplement 2.** Divergent characteristics of site-directed missense mutants of *STE2* gene.

**Figure supplement 3.** Exogenous supplementation of selective metabolites rescues α-factor-mediated programmed cell death (PCD).

complexes and observed stable complexes in the case of zymosterol (ZST) and lanosterol (LST) while observing fluctuations in the ligand RMSD for ubiquinone 6 (CoQ6) post 100 ns (*Figure 3—figure supplement 1c*), which may be a result of its greater conformational flexibility. To gain further insight into the contributing residues from the MD simulations, we performed a residue-wise decomposition analysis that provides information about the energy contributions from the different residues to total binding free energies (*Figure 3—figure supplement 2a*). These results suggest that IC4 and IC5 specific residues predominantly contribute to the total binding free energies. Notably, the binding free energies are obtained as an average of over 500 configurations corresponding to the last 50 ns of the MD simulations.

To gain deeper insight into the mode of action of these metabolites in inhibiting Ste2p signaling, we first analyzed their impact at the orthosteric site, that is, α-factor binding. We performed protein-peptide docking of Ste2p and α-factor and observed that metabolite-binding at the Ste2p–Gpa1p interface favors the α-factor interaction, as inferred from the binding free energies ΔG (kcal/mol) (*Figure 3—figure supplement 2b, c*). We also analyzed protein–protein interaction between the Ste2p (GPCR) and miniGpa1p (55) by selecting GPCR configurations with and without metabolite-induced altered cavity topologies (IC4 and IC5), respectively (frames output from the aforementioned Ste2p–metabolite complex simulations), and computed the dissociation constant ($K_d$), binding affinity (ΔG), and the structural changes in the overall Ste2p topology (*Figure 3d, e*, *Figure 3—figure supplement 2b–d*). These computational analyses revealed that, in contrast to the metabolite-free Ste2p–Gpa1p interaction, referred to as the wild-type (WT) condition, the $K_d$ value is many-fold lower in the presence of metabolites, indicating a cohesive response induced by these metabolites. A multi-fold lower $K_d$ value further indicates and potentially explains that the metabolite binding favors the Ste2p (GPCR) and miniGpa1–protein interaction and enables the establishment of a stable complex that

might influence the shielding of the effector-regulating domains of the Gpa1p or influence its binding with the Ste4p (Gβ)–Ste18p (Gγ) complex.

We next asked whether the observed resistance toward α-factor-induced PCD in single metabolic mutants might be the direct consequence of the identified metabolites or a pleiotropic response due to an altered genome. To test this, we exogenously supplemented wild-type yeast with zymosterol (ZST), ubiquinone 6 (CoQ6), and lanosterol (LST). Yeast cells were pre-loaded with metabolites for 24 hr at concentrations (0.1, 1, and 10 µM) that did not significantly alter growth profiles, except for 10 µM lanosterol, which showed mild perturbation (*Figure 3—figure supplement 3a*), thereby ensuring the physiological relevance of our experimental conditions. Following pre-treatment, cells exhibited marked rescue from α-factor-induced PCD across multiple assays, including growth kinetics, PI-based viability, and FUN1 measurements (*Figure 3f, g*, *Figure 3—figure supplement 3b, c*). This protective effect was specific to α-factor-induced PCD, as metabolite-treated cells remained sensitive to acetic acid-induced death, suggesting that it is likely due to modulation of the Ste2p–Gpa1p interface and not a consequence of an altered response (*Figure 3g*). Next, we probed whether the observed modulation of Ste2 signaling also applies to the natural yeast mating behavior. We investigated this by performing a mating assay, which also revealed a decline in Ste2 signaling in metabolite-pre-loaded cells, suggesting the interlink between these metabolites and Ste2p signaling. Notably, in the case of CoQ6, we failed to observe any significant decline in the mating response, consistent with its instability observed in MD simulations (*Figure 3h*, *Figure 3—figure supplement 3d*).

Further, we evaluated the deactivation of the pathway at the MAPK signaling level by monitoring the Fus3 phosphorylation, where the α-factor-induced p-Fus3 levels were significantly suppressed by ZST and LST (*Figure 3i*) but not CoQ6 (data not shown). For final validation, we employed a $P_{FUS1}$-eGFP-CYC1 transcriptional reporter system. Metabolite pretreatment significantly reduced eGFP-positive cells following α-factor stimulation compared to controls (*Figure 3j*), further demonstrating that the tested metabolites can inhibit α-factor-induced Ste2p signaling. These findings suggest endogenous metabolites as modulators of Ste2p signaling by stabilizing Ste2p–Gpa1p interactions.

## Site-directed *Ste2* mutants abrogate metabolite-mediated rescue phenotype

To gain a deeper understanding, we investigated the role of metabolite binding in modulating Ste2p–Gpa1p interaction dynamics and employed both computational and experimental approaches. First, we conducted in silico screening by generating site-directed mutants of Ste2p designed to alter metabolite binding, followed by docking with Gpa1p to assess the impact of these mutations. We selected mutation sites by analyzing non-covalent interactions in Ste2p–metabolite complexes (IC4 and IC5). We prioritized stronger hydrogen bonds over weaker hydrophobic interactions and pinpointed specific interaction sites for CoQ6 (S75, R233), ZST (L289), and LST (T155, V152, and I153). We finally applied a 2.7–3.3 Å ideal distance range for hydrogen bonds and selected mutants S75A for CoQ6, L289K for ZST, and T155D for LST. This filtering step eliminated unstable or distorted hydrogen bonds. These mutations significantly increased the dissociation constant ($K_d$) of the Ste2p–Gpa1p complex, indicating weakened interactions compared to wild-type Ste2p (*Figure 4a, b*, *Figure 4—figure supplement 1a*). This computational evidence supports the hypothesis that metabolite binding stabilizes the Ste2p–Gpa1p complex, facilitating the rescue response to α-factor-induced PCD.

Next, we performed experimental validation by generating site-directed missense mutants targeting key binding residues at the Ste2p–Gpa1p interface and confirmed these computational predictions (*Figure 4—figure supplement 1b*, *Supplementary file 13*). These mutants were expressed in a *ste2Δ* background, with the reconstituted wild-type *STE2* (rtWT) serving as a control. Using fluorometry-based cell death assays, it was assessed whether mutants retained the metabolite-mediated rescue response observed in wild-type cells. While rtWT exhibited significant cell death upon α-factor exposure that could be rescued by metabolite pretreatment, the T155D and L289K mutants showed no rescue response despite pretreatment with their respective metabolites (*Figure 4c*). This loss of rescue suggests direct metabolite regulation of Ste2p signaling at the intracellular Ste2p–Gpa1p interface. Notably, S75A mutants showed minimal α-factor responsiveness overall, likely due to significant structural disruptions affecting both pheromone sensitivity and metabolite binding. FUN1 staining assay and p-Fus3 signaling analysis by mapping MAPK pathway activation further supported these findings (*Figure 4d, e*). The lack of rescue highlights the direct role of metabolite binding at the Ste2p–Gpa1p

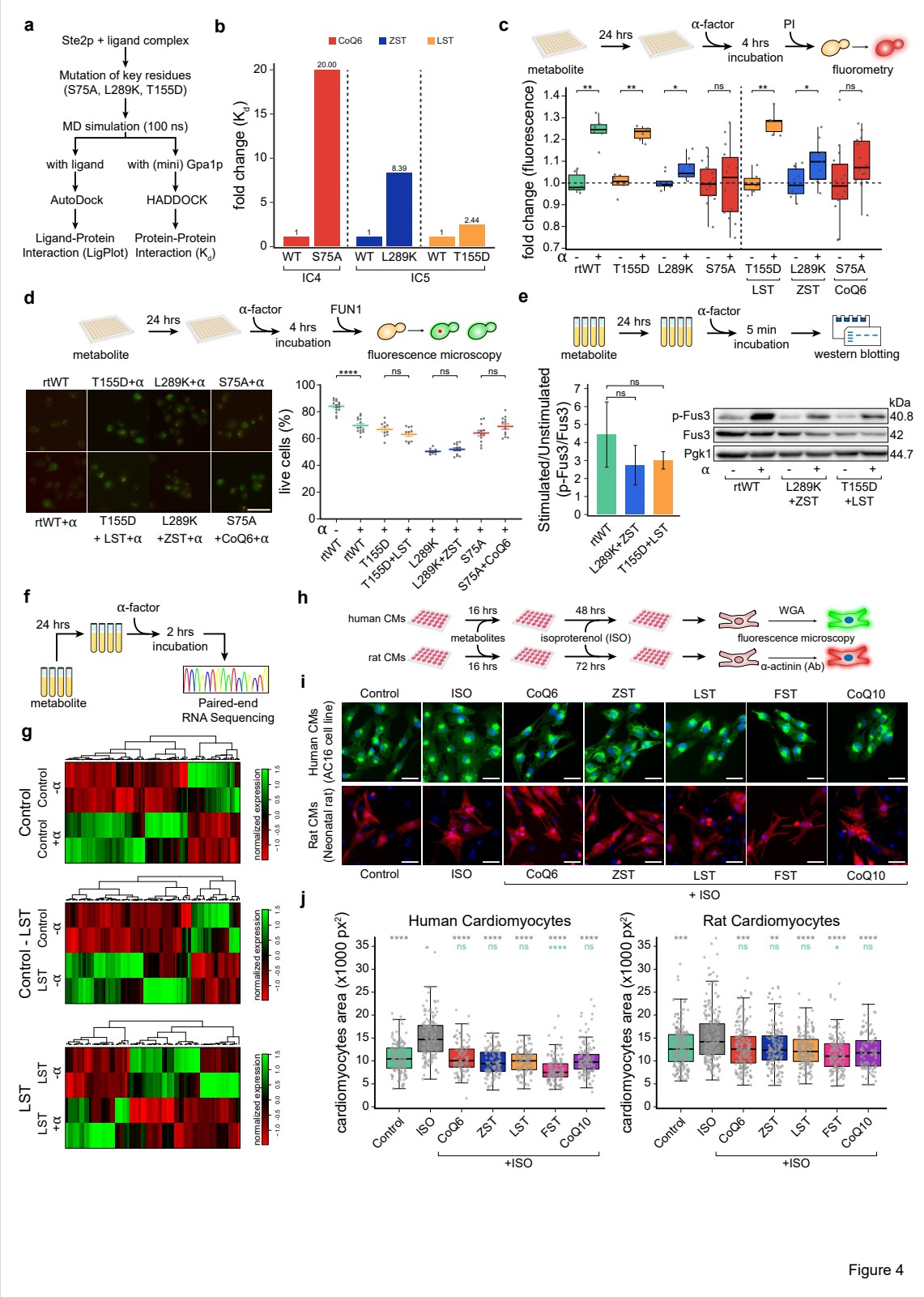

Figure 4. Site-directed Ste2p mutants disrupt metabolite-mediated rescue. (**a**) Workflow depicting the steps involved in Ste2p-miniG-protein docking of the wild-type and site-directed Ste2p mutants. Notably, docking was performed using HADDOCK and PRODIGY web servers. (**b**) Bar plots depicting the dissociation constant ($K_d$) fold change in Ste2p site-directed mutants and wild-type. Notably, fold change was computed with respect to the metabolite-influenced wild-type condition (Ste2p-miniG-protein). (**c**) The schematic diagram depicts the experimental workflow used to quantify α-factor-induced

*Figure 4 continued on next page*

*Figure 4 continued*

programmed cell death in generated site-directed missense mutants (T155D, L289K, and S75A), alongside reconstituted wild-type *STE2* (rtWT), using a propidium iodide-based cell viability fluorometric assay. The box plot (left) depicts the increase in the relative proportion of dead cells upon α-factor exposure. Box plot (right) depicting the loss of rescue phenotype from the α-factor-induced programmed cell death in the indicated conditions when pre-loaded with metabolites as inferred using propidium iodide-based cell viability fluorometric assay. The y-axis represents the fold change of the propidium iodide fluorescence values with respect to their respective controls. The Mann–Whitney *U* test was used to calculate statistical significance. Asterisks indicate statistical significance, whereas ns represents non-significance. (**d**) Schematic representation (top) of the experimental approach used to measure cell vitality and viability using microscopy-based FUN1 staining. Representative micrographs (below) depicting the FUN1 staining results in the indicated conditions, Scale 10 µm. Mean-whisker plot depicting the relative proportion of the vital and viable yeast cells observed using FUN1 staining in the indicated conditions (*n* = 4 biological replicates). A Student's *t*-test was used to compute statistical significance. Asterisks indicate statistical significance, whereas ns represents non-significance. Error bars represent the standard error of the mean (SEM). (**e**) Schematic representation (up) depicting the experimental design of phospho-MAPK activity-based Western blot. Bar plots (down) depicting the p-Fus3 levels (mean ± SEM) in the indicated conditions (n = 3 biological replicates). The y-axis represents the p-Fus3/Fus3 ratio for the stimulated condition normalized by its corresponding unstimulated sample. A Student's *t*-test was used to compute statistical significance. Asterisks indicate statistical significance, whereas ns represents non-significance. (**f**) Schematic representation depicting the experimental design of RNA sequencing, featuring treatment duration and the sequencing parameters. (**g**) Heatmap depicting the expression of differentially expressed genes obtained from RNA sequencing in the indicated conditions. Notably, Control and LST represent yeast cells unloaded and pre-loaded with lanosterol, respectively. α-factor is represented as α, where plus and minus signs represent its presence and absence, respectively. (**h**) Schematic representation of the experimental workflow followed to deduce the impact of indicated metabolites treatment on isoproterenol (ISO)-induced, GPCR-mediated hypertrophy response in human (AC16) and neonatal rat cardiomyocytes. Notably, in the case of AC16 cells, wheat germ agglutinin (WGA) was used to stain the cardiomyocytes, whereas, for neonatal cardiomyocytes, alpha-sarcomeric actinin staining was used. (**i**) Micrographs depicting the human (above; green colored) and neonatal rat (below; red colored) cardiomyocytes in the indicated conditions. Scale 50 µm. (**j**) Box plots depicting the surface area of human (AC16) and neonatal rat cardiomyocytes in the indicated conditions. Statistical significance of indicated metabolites with untreated control and isoproterenol-treated conditions is indicated in green and gray text, respectively. Mann–Whitney *U* test with Bonferroni-corrected p-values was used to compute statistical significance.

The online version of this article includes the following source data and figure supplement(s) for figure 4:

**Source data 1.** PDF file containing original western blots of p-Fus3, Fus3, and Pgk1 displayed in *Figure 4e*.

**Source data 2.** Original files for western blot analysis of p-Fus3, Fus3, and Pgk1 displayed in *Figure 4e*.

**Figure supplement 1.** Effects of Ste2 mutants on shmoo formation in response to α-factor.

**Figure supplement 2.** RNA-sequencing unveils genes involved in attenuating α-factor-induced cell death response.

**Figure supplement 3.** Evolutionary conservation of GPCR–Gα interface.

interface in regulating downstream signaling. Interestingly, S75A mutants showed no α-factor-induced effects, likely due to significant structural disruptions. Shmoo formation assays further corroborated these findings, with no rescue effects observed in the mutants despite metabolite pre-loading (*Figure 4—figure supplement 1c–e*).

To gain an unbiased view of the mode of action of these metabolites in attenuating Ste2p (GPCR)-mediated pheromone-induced cell death in yeast, we performed RNA sequencing on LST-pre-loaded and untreated (control) cells with and without α-factor exposure (*Figure 4f*). Transcriptomic analysis of control cells (without metabolite treatment) revealed significant expression changes in genes related to critical cellular processes upon α-factor exposure (*Figure 4g*, top panel; *Figure 4—figure supplement 2c*). A more detailed and careful examination revealed differential expression of genes implicated in PCD and mating responses, including *GSY1*, whose downregulation was linked to α-factor resistance (*Figure 4f, g*, *Figure 4—figure supplement 2a–c*, *Supplementary files 7–10*). LST treatment alone (without α-factor) also induced significant differential gene expression associated with cellular processing, validating its bioactivity (*Figure 4g*, middle panel). However, comparison of LST-pre-loaded versus control cells following α-factor treatment revealed no prominent mating or PCD-related genes among the differentially expressed genes (*Figure 4g*, lower panel); however, one cannot rule out the contribution of these genes in providing innate resistance to the α-factor. We tested the contribution of these genes in facilitating metabolite-mediated rescue phenotypes using the α-factor-induced cell-death assay on the knockouts of these differentially upregulated transcripts. Our results showed a significant loss of metabolite-mediated rescue phenotype in 6 out of 10 knockouts, with *YCR095W-A* displaying the most pronounced phenotype loss (*Figure 4—figure supplement 2d–f*). These results collectively suggest that metabolite binding at the Ste2p–Gpa1p interface directly drives rescue responses, with secondary contributions from differentially expressed genes in attenuating α-factor-induced cell death.

Since our intensive computational interrogation of all the available human GPCR–Gα complexes revealed a higher degree of functional conservation (*Figure 1k*), we next explored whether intracellular allosteric modulators such as ubiquinone 6 (CoQ6), zymosterol (ZST), lanosterol (LST), fucosterol (FST), and ubiquinone 10 (CoQ10) modulate GPCR signaling in higher vertebrates such as human and rat beta 1/2-adrenergic receptors signaling. Briefly, by using Gcoupler, we identified the putative GPCR–Gα interface and performed molecular docking with the aforementioned metabolites. Docking results revealed a high binding affinity of these selected metabolites at the GPCR–Gα interface of adrenergic receptors, reminiscent of Ste2p–metabolite interactions (*Figure 4—figure supplement 3a, b*). Sequence conservation analysis of the GPCR–Gα interface across yeast Ste2p and adrenergic receptors in humans and rats further confirmed a high degree of evolutionary conservation at the metabolite-binding residues (*Figure 4—figure supplement 3c*). Finally, to test this functional relevance, we evaluated the effect of these metabolites on isoproterenol-induced adrenergic receptor-mediated cardiac hypertrophy in human AC16 cardiomyocytes and neonatal rat cardiomyocytes. Pre-loading cells with these metabolites significantly attenuated hypertrophic responses, as evidenced by reduced single-cell surface area in quantitative assessments (*Figure 4h–j*). Notably, to further evaluate the evolutionary conservation of this phenomenon, we also analyzed 75 unique GPCR–Gα complex structures from six species, selected from the PDB database. Dynamic docking was performed using five metabolites (CoQ6, ZST, LST, FST, and CoQ10) identified by Gcoupler as potential allosteric modulators and five negative controls predicted as poor binders. Results revealed significantly lower docking scores ($<-7$ kcal/mol) for Gcoupler-recommended metabolites compared to negative controls, irrespective of GPCR type or species (*Figure 4—figure supplement 3d–h*). These findings demonstrate that intracellular metabolite modulation of GPCR activity is a conserved mechanism extending beyond yeast to higher vertebrates.

## Discussion

Over the last few decades, extensive research has focused on identifying allosteric modulators of GPCRs due to their relevance in drug discovery (*Leach et al., 2007*; *Topiol, 2018a*). Most known modulators are exogenous and target extracellular sites, while intracellular allosteric sites, identified recently through structural biology, offer novel avenues for regulation (*Calebiro et al., 2021*; *Rees et al., 2002*; *Wold and Zhou, 2018*; *Leach et al., 2007*; *van der Westhuizen et al., 2015*; *Topiol, 2018a*). These sites, overlapping with G-protein and β-arrestin coupling regions, highlight the potential for intracellular allosteric modulation (*Bourque et al., 2022*; *van der Westhuizen et al., 2015*; *Yang et al., 2021*). Intracellular modulators, including chemically diverse agents like auto-antibodies and sodium ions, remain poorly understood, emphasizing the need for systematic exploration of these sites (*Wold and Zhou, 2018*; *Bourque et al., 2022*; *Rees et al., 2002*; *van der Westhuizen et al., 2015*; *Stornaiuolo et al., 2015*; *Reyes-Alcaraz et al., 2020*; *Nachtergaele et al., 2012*; *Zhang et al., 2015*; *O'Callaghan et al., 2012*). However, the lack of data on intracellular modulators limits the feasibility of conventional computational approaches (*Basith et al., 2018*; *Congreve et al., 2020*; *Hedderich et al., 2022*; *Chatzigoulas and Cournia, 2021*; *Bartuzi et al., 2018*; *Hou et al., 2021*; *Topiol, 2018b*).

To address this gap, we developed Gcoupler, a computational framework integrating de novo cavity identification, ligand synthesis, statistical analysis, graph neural networks, and bioactivity-based ligand prioritization. Unlike existing tools, Gcoupler does not require cavity-specific experimentally validated compounds for model training. Gcoupler's precision in cavity mapping, flexibility for user-defined queries, and ability to screen large chemical libraries make it a versatile and efficient tool. Additionally, Gcoupler's generic design allows application beyond GPCRs, contrasting with existing platforms that often have limitations in modularity, precision, or open-source availability. Noteworthy, in contrast to other known allosteric sites identification tools for GPCRs, such as Allosite (*Huang et al., 2013*), AllositePro (*Song et al., 2017*), AlloReverse (*Zha et al., 2023*), that largely leverage the ML-based models or require orthosteric ligand-bound structure as input, the cavity detection feature of Gcoupler (LigBuilderV3), a critical step for the entire workflow, is not limited to only the allosteric sites; instead, it identifies all possible cavity-like regions on the protein surface, which then get classified into druggable, undruggable, or amphibious based on their individual scoring and ligandability, thus making it unbiased and more specific toward query protein. Notably, the rationale for opting for LigBuilder V3 for cavity identification over similar tools such as Fpocket (*Le Guilloux et al., 2009*)

is that the former uses a hydrogen atom probe, moving along the protein surface grid of 0.5 Å for cavity detection, being much more precise in detecting cavity boundaries, in both breadth and depth mapping; in contrast, the latter considers clusters of alpha spheres (*Supplementary file 11*).

To date, only a few methods leverage generative AI models for cavity/pocket-based drug design. Gcoupler is an open-source, end-to-end platform integrating Ligand-Based Drug Design (LBDD) and Structure-Based Drug Design (SBDD) for drug design and large-scale screening. Unlike Pocket Crafter (*Shen et al., 2024*), which requires proprietary tools (e.g., MOE QuickPrep) and lacks predictive model-building modules, Gcoupler offers comprehensive functionality. Similarly, DeepLigBuilder (*Li et al., 2021*) and Schrodinger's AutoDesigner are either closed source or limited in features compared to Gcoupler. Comparative analysis highlights Gcoupler's unique advantages in precision, flexibility, and functionality (*Supplementary file 11*).

Using Gcoupler, we investigated the molecular basis of innate resistance to α-factor-induced PCD in yeast. Unlike humans, yeast possess only two GPCR systems, making their pheromone-sensing pathway ideal for focused study. Previous research predominantly identified downstream regulatory mechanisms (*Velazahan et al., 2021*; *Sokolov et al., 2020*; *Alvaro and Thorner, 2016*); however, our findings suggest an upstream, receptor-level regulation via endogenous intracellular metabolites. Computational and experimental evidence pinpointed specific metabolites binding to the Ste2p–miniGpa1 interface, modulating signaling. Site-directed mutagenesis confirmed the functional relevance of these metabolite-interacting residues. Of note, previous mutagenesis experiments also revealed multiple critical amino acid residues that overlap with IC4 and IC5, suggesting their functional relevance in Ste2p downstream signaling (*Supplementary file 12*). Despite these advances, certain limitations remain, including the potential pleiotropic effects in metabolic gene knockouts and challenges in replicating natural metabolite concentrations. Mechanistic insights into sterol biosynthesis mutants (*ergΔ*) revealed impaired mating responses due to heterogeneous defects, such as reduced sterol accumulation and shmoo formation, impaired membrane fusion, and decreased *FUS1* expression (*Bagnat and Simons, 2002*; *Jin et al., 2008*; *Heese-Peck et al., 2002*; *Aguilar et al., 2010*; *Tiedje et al., 2007*). This highlights a novel role for sterols in GPCR regulation and their broader implications for yeast microbial factories and stress tolerance (*Shaw et al., 2019*; *Ostrov et al., 2017*). Sterols and other endogenous metabolites were shown to modulate GPCR activity by targeting conserved Gα-binding sites, reinforcing the evolutionary conservation of this mechanism across species, as demonstrated in human and rat hypertrophy models in this study.

In summary, our work uncovers a novel regulatory mechanism for GPCRs mediated by intracellular metabolites and presents a computational framework, Gcoupler, to explore unexplored allosteric sites. The proposed model suggests that selective metabolites binding to GPCR–Gα interfaces induce local conformational changes, stabilizing GPCR–G-protein complexes and potentially obstructing downstream signaling. Alternative mechanisms, such as orthosteric site modulation, kinase/arrestin interaction interference, or alterations in membrane dynamics, remain to be explored, warranting further investigation. A critical limitation of our study is the absence of direct binding assays to validate the interaction between the metabolites and Ste2p. While our results from genetic interventions, MD simulations, and docking studies strongly suggest that the metabolites interact with the Ste2p–Gpa1 interface, these findings remain indirect. Direct binding confirmation through techniques such as surface plasmon resonance, isothermal titration calorimetry, or co-crystallization would provide definitive evidence of this interaction. Addressing this limitation in future work would significantly strengthen our conclusions and provide deeper insights into the precise molecular mechanisms underlying the observed phenotypic effects. Another critical limitation of our findings is its reliance on tools like AutoDock and PRODIGY for preliminary binding affinity estimates, which lack the thermodynamic precision of advanced methods. Although calibrating docking scores with experimental data or using alchemical free energy calculations can improve accuracy, these methods are computationally expensive and require high-quality data, which is often unavailable. Gcoupler prioritizes speed, scalability, and accessibility, especially for data-sparse scenarios. By focusing on efficient, data-driven classification methods, it balances performance with practicality for large-scale screening. In this study, to address this limitation, we employed MD simulations with molecular mechanics-Generalizer Born surface area approach, incorporating factors like protein flexibility and solvation effects for more accurate $\Delta G$ calculations. While computationally intensive approaches were beyond this study's scope, we ensured reported $\Delta G$ values reflected system conformational flexibility by basing them on

pre-simulated docked structures from MD simulations. Further, our results suggest that the metabolite binds to the Ste2p–Gpa1 interface and modulates receptor activity upon pheromone stimulation, as supported by various assays. However, the precise sequence of interactions between Ste2p, the metabolite, and Gpa1 remains unexplored, as it requires sequential experiments beyond this study's scope. Taken together, addressing these limitations in future work would significantly strengthen our conclusions and provide deeper insights into the precise molecular mechanisms underlying the observed phenotypic effects.

## Materials and methods
### Backend code for the Gcoupler
The back-end code for Gcoupler is implemented entirely in Python (3.8) and comprises four modules: Synthesizer, Authenticator, Generator, and BioRanker. Synthesizer employs LigBuilder V3.0 (*Yuan et al., 2020*) for de novo in silico ligand synthesis, identifying protein cavities likely to be active or allosteric sites using a hybrid GROW-LINK approach with a Genetic Algorithm. The module autonomously selects one cavity for ligand synthesis based on user-defined residue positions. Using the CAVITY function of LigBuilder (*Yuan et al., 2011*), it classifies 3D grid points around the protein into occupied, vacant, and surface points and integrates geometric and physicochemical properties to identify binding sites (*Clark et al., 1989*). Synthesizer outputs ligand structures in SMILES and PDBQT formats, alongside cavity grid coordinates for downstream modules. Authenticator validates the synthesized ligands using AutoDock Vina (1.2.3) for virtual screening (*Trott and Olson, 2010*). Binding energy calculations classify ligands into HABs and LABs, preserving balance for subsequent deep learning analysis. The default energy threshold is set to –7 kcal/mol, but users can explore alternative cutoffs, visualize distributions, and perform statistical comparisons within the workflow. These steps ensure precise identification and prioritization of ligand candidates for further analysis. The Authenticator uses the Kolmogorov–Smirnov test (*Berger and Zhou, 2014*), Anderson–Darling test (*Engmann and Cousineau, 2011*), and Epps–Singleton test (*Goerg and Kaiser, 2009*) for hypothesis testing for the comparison of the distributions. The Authenticator module visualizes ligand distributions using overlapping density plots and ECDF curves. If the default threshold fails to produce statistically meaningful separation, users can supply alternative negative datasets, such as decoys generated via Gcoupler's inbuilt RDKit Chem module or custom datasets (*Landrum, 2023*). The Generator module builds deep learning-based classification models using the DeepChem (2.6.1) library (*Ramsundar et al., 2019*). It accepts HABs and LABs/decoys from the Authenticator module to train four graph-based models: GCM, GCN (*Kipf and Welling, 2016*), AFP (*Xiong et al., 2020*), and GAT (*Veličković et al., 2018*). Class imbalance is addressed through upsampling techniques. The generator tests all models using default hyperparameters, returning performance metrics for user selection. Hyperparameters can be tuned via manual settings or using default values followed by $k$-fold cross-validation. The final optimized model, trained on the complete synthetic dataset (HAB + LAB/decoys), enables large-scale screening of user-supplied compounds based on their SMILES representations. The BioRanker module performs post-prediction analysis for functional activity-based compound screening. Positively predicted compounds are selected using a stringent probability threshold or adaptive methods such as $G$-means and Youden's $J$ statistic, which optimize sensitivity and specificity. The selected compounds are projected into biological activity spaces (Chemistry, Targets, Networks, Cells, and Clinics) by comparing their biological activity descriptor vectors with those of HABs using cosine similarity (*Bertoni et al., 2021*). A modified PageRank algorithm ranks compounds based on activity-specific scores, with support for multi-activity ranking to refine results based on user-defined biological properties, ensuring precise and context-relevant compound prioritization.

Additional information about the backend code for Gcoupler, along with methodology for runtime analysis, sequence–structural–functional level analysis, MD simulation, molecular docking (AutoDock), functional enrichment analysis, and protein–protein docking, can be accessed in Appendix 1.

### Gcoupler benchmarking
To assess batch effects across Gcoupler runs for a specific cavity, we utilized the standard Gcoupler Docker image. Intracellular cavity 4 (IC4) of the Ste2 protein of yeast was used for benchmarking. A total of 100 molecules were in silico synthesized by the Synthesizer module of Gcoupler iteratively.

Post-generation, atom pair fingerprints (ChemmineR; R package) were calculated for the synthesized molecules from each run, and the data was visualized using principal component analysis and pairwise comparison using Tanimoto Similarity (ChemmineR, R package).

For model benchmarking, Gcoupler was validated on GPCRs from the DUD-E dataset, alongside the information about the active ligands and their randomly selected number-matched decoys (*Mysinger et al., 2012*). Additionally, Gcoupler's performance in identifying experimentally elucidated allosteric sites and modulators was tested using PDB complexes obtained from the RCSB PDB database.

## Metabolomics

Wildtype (BY4741) and *ste2Δ* yeast strains were grown in YPD medium at 30°C, 150 rpm, through primary and secondary cultures (16 hr each). Equal cell numbers (1.5 ml) were aliquoted into a 96-well deep well plate. α-factor (Sigma-Aldrich) was added at final concentrations of 10, 20, 30, 40, and 50 µM (eight replicates each). DMSO served as the solvent control, while untreated WT and *ste2Δ* conditions received no treatment. Plates were incubated (30°C, 150 rpm, 4 hr) under a breathable membrane. A 50 µl aliquot was taken for PI (11195, SRL) assay as described in Appendix 1. Following the PI assay, four pooled replicates were pelleted (6000 rpm, 5 min, RT), treated with zymolyase (40 U/ml, 1X PBS, 30°C, 1 hr), washed with PBS, and metabolomics analysis was performed. Data analysis included peak normalization, omission of metabolites with constant or >50% missing values, and kNN-based imputation (MetaboAnalyst). Data were interquartile range-filtered, and differentially enriched metabolites were identified by calculating $\log_2$ fold change ($|\log_2FC| \geq 1$, $p < 0.05$ via Student's *t*-test). Pathway Over-Representation Analysis was performed using MetaboAnalyst with hypergeometric or Fisher's exact tests to assess pathway enrichment against background metabolite distributions. Further details about the methodology are available in Appendix 1.

## Genetic screening

Fifty-three knockout strains from the Yeast Deletion Collection, along with WT and *ste2Δ*, were treated with α-factor (30 µM), while DMSO served as the solvent control. Plates were incubated for 4 hr. A 50 µl aliquot was used to measure the PI-based cell viability assessment assay as described in Appendix 1. Fluorescence data were normalized to blank-adjusted $OD_{600}$, followed by two additional rounds of normalization with unstained and HK controls. The percentage fold change for the treated group was calculated relative to the untreated group, and statistical significance was determined using a one-sample Student's *t*-test. Further details about the methodology used are available in Appendix 1.

## Pre-loading of yeast cells with a metabolite

Yeast cells were cultured in YPD medium at 30°C, 200 rpm for 16 hr in primary and secondary cultures. Equal cell densities (5 µl) from secondary cultures were inoculated into 96-well plates containing 145 µl YPD with metabolites coenzyme Q6 (CoQ6, 900150O, Avanti Polar Lipids), zymosterol (ZST, 700068P, Avanti Polar Lipids), and lanosterol (LST, L5768, Sigma-Aldrich) at 0.1, 1, and 10 µM concentrations. Plates were incubated for 24 hr at 30°C, 200 rpm, with multiple biological replicates. Ethanol-treated wells served as solvent controls. For site-directed *STE2* mutants, the mutants were grown in YPD for primary and secondary cultures, but the metabolite pre-loading was performed in YPGR instead of YPD to induce Ste2 expression.

After pre-loading, the following assays were performed: growth kinetics, PI-based assay, FUN1 staining, mating assay, phospho-MAPK activity-based western blot, and transgenic reporter assay. The detailed protocol for each of these assays is available in Appendix 1.

## RNA-sequencing

Yeast cells (BY4741) were cultured in YPD medium at 30°C, 200 rpm, with or without lanosterol (LST, 1 µM) in biological duplicates. Cells were subsequently treated or untreated with α-factor (30 µM) for 2 hr. RNA was isolated following *Mittal et al., 2022*. Sequencing quality was assessed using MultiFastQ, and paired-end reads were trimmed and aligned to the *S. cerevisiae* reference genome (ENSEMBL R64-1-1; GCA_000146045.2) using the Rsubread package (v2.6.4). Gene expression counts were generated via featureCounts and normalized using TMM (*Supplementary file 7*). Differential expression analysis was performed with NOISeq (v2.38.0) (*Supplementary files 8–10*). Functional

enrichment was assessed using the Gene Ontology Term Finder (v0.86) from the *Saccharomyces* Genome Database. Raw FASTQ files and normalized expression data are available on Zenodo. Additional details are available in Appendix 1.

### Site-directed mutagenesis

The gene encoding wild-type *STE2* was PCR amplified from the genome of *S. cerevisiae* and further cloned into plasmid pRS304 under galactose-inducible *GAL1* promoter to generate plasmid pRS304-$P_{GAL1}$-STE2-CYC1 using Gibson assembly (*Gibson et al., 2010*; *Gibson et al., 2009*). The mutants were generated by PCR amplifying the gene with primers consisting of respective mutations and cloned into plasmid pRS304 under *GAL1* promoter to generate pRS304-$P_{GAL1}$-STE2 S75A-CYC1, pRS304-$P_{GAL1}$-STE2 T155D-CYC1, and pRS304-$P_{GAL1}$-STE2 L289K-CYC1. Wild-type (rtWT) and mutant *STE2* were integrated by digesting the pRS304 vector with restriction enzyme BstXI to generate a linearized plasmid and transformed into the *S. cerevisiae* BY4741 *ste2Δ* strain. Additional details about the methodology are available in **Appendix 1**.

### Cardiomyocyte hypertrophy models

Human AC16 cardiomyocytes were cultured in DMEM-F12 (Thermo Scientific) with 12.5% fetal bovine serum (FBS) at 37°C and 5% $CO_2$. Cells were seeded in a 24-well plate for size measurements, treated after 24 hr with metabolites (CoQ6, ZST, LST, FST, and CoQ10) at 2.5 μM, and incubated overnight with 1% FBS. The medium was refreshed with fresh metabolites and isoproterenol (25 μM) for 48 hr. Cells were washed with PBS, fixed with 4% paraformaldehyde, and stained with wheat germ agglutinin (Thermo Scientific) and DAPI. Images were captured using a Leica DMI 6000 B microscope at 20× magnification, and cell area was measured using ImageJ. Neonatal rat cardiomyocytes were isolated from 1- to 3-day-old SD rat pups using Collagenase Type II. After heart explantation and digestion, the cells were centrifuged and pre-plated for 90 min to remove fibroblasts. The cardiomyocytes were seeded in a gelatin-coated 24-well plate, incubated overnight with 2.5 μM metabolites and 1% FBS, and then treated with metabolites (2.5 μM) and isoproterenol (10 μM) for 72 hr. Cells were fixed, stained with alpha-sarcomeric actinin and DAPI, and images were captured using a Leica DMI 6000 B at 20× magnification. Cell area was quantified using ImageJ. Additional details about the methodology are available in Appendix 1.

### Statistical analysis

Statistical analyses were performed using Past4 software or R Programming. The Mann–Whitney *U* test was applied to compare medians between two distributions (non-parametric), while Student's *t*-test was used for pairwise comparisons of means. p-value correction was performed using the Bonferroni method when necessary. A significance threshold of 0.05 was set, with *, **, ***, and **** indicating p-values <0.05, <0.01, <0.001, and <0.0001, respectively.

### Materials availability

All yeast strains used in this study are available from the corresponding author upon reasonable request.

## Acknowledgements

The authors thank the IT-HelpDesk team of IIIT-Delhi for assisting with the computational resources. We thank all the members of the Ahuja lab for their intellectual contributions at various stages of this project. We thank Prof. G.P.S. Raghava for providing critical comments on our manuscript. We thank Dr. Arjun Ray for providing intellectual support. We also thank Dr. Martin Graef and Dr. Kaushik Chakraborty for sharing yeast strains and the NIPER Guwahati central facility for helping us with high-resolution metabolomics. The Ahuja lab is supported by the Ramalingaswami Re-entry Fellowship (BT/HRD/35/02/2006), a re-entry scheme of the Department of Biotechnology, Ministry of Science & Technology, Government of India, Start-Up Research Grant (SRG/2020/000232) from the Science and Engineering Research Board, and a research grant from IHUB Anubhuti (Project Grant/23) and an intramural Start-up grant from Indraprastha Institute of Information Technology-Delhi. The INSPIRE faculty grant from the Department of Science & Technology, India, funds the Sengupta lab. Gupta Lab

is funded by the Ramalingaswami Re-entry Fellowship (BT/RLF/re-entry/14/2019) from the Department of Biotechnology, Government of India.

## Additional information

### Funding

| Funder | Grant reference number | Author |
|---|---|---|
| Department of Biotechnology, Ministry of Science and Technology, India | BT/HRD/35/02/2006 | Gaurav Ahuja |
| Science and Engineering Research Board | SRG/2020/000232 | Gaurav Ahuja |
| Department of Science and Technology, Ministry of Science and Technology, India | BT/RLF/re-entry/14/2019 | Gaurav Ahuja |
| IHUB Anubhuti | IHUB Anubhuti/Project Grant/23 | Gaurav Ahuja |

The funders had no role in study design, data collection, and interpretation, or the decision to submit the work for publication.

### Author contributions

Sanjay Kumar Mohanty, Conceptualization, Data curation, Software, Formal analysis, Validation, Visualization, Methodology, Project administration, Writing – review and editing; Aayushi Mittal, Formal analysis, Validation, Visualization, Methodology, Writing – review and editing; Namra Farooqi, Data curation, Validation; Aakash Gaur, Subhadeep Duari, Saveena Solanki, Anmol Kumar Sharma, Suvendu Kumar, Vishakha Gautam, Nilesh Kumar Dixit, Karthika Subramanian, Tarini Shankar Ghosh, Formal analysis; Sakshi Arora, Writing – review and editing; Debarka Sengupta, Supervision; Shashi Kumar Gupta, Formal analysis, Supervision, Validation, Methodology; Arul Natarajan Murugan, Formal analysis, Supervision, Methodology; Deepak Sharma, Conceptualization, Formal analysis, Supervision, Funding acquisition, Methodology, Writing – review and editing; Gaurav Ahuja, Conceptualization, Supervision, Funding acquisition, Methodology, Writing – original draft, Writing – review and editing

### Author ORCIDs

Sanjay Kumar Mohanty ⓘ https://orcid.org/0000-0002-1375-2223
Aayushi Mittal ⓘ https://orcid.org/0000-0002-1973-8553
Namra Farooqi ⓘ https://orcid.org/0009-0000-8650-0249
Subhadeep Duari ⓘ https://orcid.org/0000-0001-9805-7635
Saveena Solanki ⓘ https://orcid.org/0000-0003-1079-9349
Anmol Kumar Sharma ⓘ https://orcid.org/0009-0007-2860-9407
Sakshi Arora ⓘ https://orcid.org/0000-0001-7535-3597
Suvendu Kumar ⓘ https://orcid.org/0000-0002-4405-8402
Vishakha Gautam ⓘ https://orcid.org/0000-0002-4755-0291
Debarka Sengupta ⓘ https://orcid.org/0000-0002-6353-5411
Deepak Sharma ⓘ https://orcid.org/0000-0003-1104-575X
Gaurav Ahuja ⓘ https://orcid.org/0000-0002-2837-9361

### Ethics

The local IAEC (Institutional Animal Ethics Committee) committee at CSIR-Central Drug Research Institute approved all the animal experiments (IAEC/2020/38) following the guidelines of the Committee for the Purpose of Control and Supervision of Experiments on Animals (CPCSEA), New Delhi, Government of India.

Reviewer #1 (Public review): https://doi.org/10.7554/eLife.106397.3.sa1

Reviewer #1 (Public review): https://doi.org/10.7554/eLife.106397.3.sa2

Author response https://doi.org/10.7554/eLife.106397.3.sa3

# Additional files

## Supplementary files

Supplementary file 1. The table contains information about the structure of the human GPCR–Gα-protein complexes downloaded from the Protein Data Bank (PDB). It also contains information about the PDB IDs, protein name, species information, whether it is used or not, UniProt IDs, and other information.

Supplementary file 2. The table contains information about the normalized root mean square deviation (RMSD) of the pairwise comparison of the GPCR–Gα-protein interface (cavities) for the indicated GPCRs.

Supplementary file 3. The table contains information about the chemical similarities of the de novo synthesized synthetic compounds of the indicated GPCR–Gα-protein interface (cavities). Of note, the chemical similarity was computed using the Tanimoto coefficient on the Atomic fingerprints.

Supplementary file 4. The table contains information about the YMDB metabolites predicted to bind at the intracellular cavities IC4 and IC5 of the Ste2 protein. Of note, predictions were made using molecular docking and Gcoupler. It contains information about the YMDB ID, metabolite name, SMILES (Simplified Molecular Input Line Entry System), and cavity information. Of note, BE refers to binding energies, whereas probabilities were the outcome of Gcoupler.

Supplementary file 5. The table contains information about the list of metabolite-gene pairs from the Kyoto Encyclopedia of Genes and Genomes (KEGG) pathway.

Supplementary file 6. The table contains the results of the untargeted metabolomics. Of note, the values represent normalized and imputed peak intensities of the indicated metabolites.

Supplementary file 7. The table contains the TMM normalized read counts after RNA sequencing.

Supplementary file 8. The table contains the list of differentially expressed genes (DEGs) of the Control groups, alongside the other standard output from the NOISeq analysis, such as M and D values, mean values of the two conditions, probability ranking, and other gene-associated features.

Supplementary file 9. The table contains the list of differentially expressed genes (DEGs) of the Control versus LST group, alongside the other standard output from the NOISeq analysis, such as M and D values, mean values of the two conditions, probability ranking, and other gene-associated features.

Supplementary file 10. The table contains the list of differentially expressed genes (DEGs) of the LST group, alongside the other standard output from the NOISeq analysis, such as M and D values, mean values of the two conditions, probability ranking, and other gene-associated features.

Supplementary file 11. The table contains information about the comparison of de novo drug design and cavity detection tools. Please note, SBDD refers to Structure-Based Drug Design, and LBDD refers to Ligand-Based Drug Design.

Supplementary file 12. The table contains information about the known mutants from the literature that could modulate Ste2 signaling. Of note, the table also contains information about the site of mutation, the nature of the mutation, its position within the intracellular cavities IC4 and IC5, its impact on the Ste2 signaling pathway, and references.

Supplementary file 13. Details of the nucleotide and amino acid Multiple Sequence Alignments (MSAs) of the wild-type STE2 and the site-directed missense mutants of STE2. Each row starts with the sequence identifier followed by the aligned sequence (in chunks) with the ending position of the aligned sequence provided at the end, separated by tabs. The mutation sites are highlighted in red (S75A), yellow (T155D), and blue (L289K).

MDAR checklist

## Data availability

The processed untargeted metabolomics data is provided as Supplementary Information. The raw RNA sequencing files are available at ArrayExpress under accession E-MTAB-12992. A Python package for Gcoupler is provided via pip https://test.pypi.org/project/Gcoupler/. A dockerDocker container pre-compiled with Gcoupler and all of its dependencies can be found at https://hub.docker.

com/r/sanjayk741/gcoupler. The source code of Gcoupler is available on the project GitHub page: https://github.com/the-ahuja-lab/Gcoupler (copy archived at Zenodo with DOI: https://doi.org/10.5281/zenodo.7835335), whereas the raw sequencing files can be accessed using DOI: https://doi.org/10.5281/zenodo.7834294.

The following datasets were generated:

| Author(s) | Year | Dataset title | Dataset URL | Database and Identifier |
| --- | --- | --- | --- | --- |
| Ahuja G | 2025 | RNA-seq of *S. cerevisiae* (BY4741) treated with Lanosterol against untreated controls in the presence and absence of alpha-factor | https://www.ebi.ac.uk/biostudies/ArrayExpress/studies/E-MTAB-12992 | Array Express, E-MTAB-12992 |
| the-ahuja-lab | 2023 | Gcoupler supplement data | https://doi.org/10.5281/zenodo.7834294 | Zenodo, 10.5281/zenodo.7834294 |
| the-ahuja-lab | 2023 | the-ahuja-lab/Gcoupler: Gcoupler | https://doi.org/10.5281/zenodo.7835335 | Zenodo, 10.5281/zenodo.7835335 |

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

## Appendix 1

### Materials and methods

Backend code for the Gcoupler

The back-end code for the Gcoupler is written entirely in Python (3.8). The Gcoupler workflow consists of four distinct modules, namely, Synthesizer, Authenticator, Generator, and BioRanker. The Synthesizer module leverages the LigBuilder V3.0 (*Yuan et al., 2020*) to identify the putative cavities on the protein surface with the potential to be an active or an allosteric site and perform de novo in silico drug synthesis within the target cavity by following a hybrid approach of GROW and LINK method using Genetic Algorithm. Notably, among all the predicted cavities, the Synthesizer module pinpoints one cavity as the target for the subsequent in silico ligand synthesis process. The decision is autonomous in nature and is taken based on user-supplied residue-position pairs of interest. The Synthesizer module employs the CAVITY (a structure-based protein binding site detection program) function of LigBuilder V3 for cavity detection (*Yuan et al., 2011*). It creates 3D grid points at a cushion of 0.5 Å to contain the whole protein. It uses a Tripos force field (*Clark et al., 1989*) to categorize these points into three groups: 'occupied grid points', which contain protein atoms or lie within the range of solvent-accessible radii; 'vacant grid points', located outside the protein atoms; and 'surface grid points', positioned between the occupied and vacant grid points. Following this classification, a geometric approach is employed to identify potential sites accessible to ligands, also influenced by factors such as depth and volume. Finally, both geometric structural information and physical chemistry properties are utilized to pinpoint locations for ligand binding sites and to quantitatively assess the feasibility of drug binding within each potential site. Gcoupler identifies multiple potentially orthosteric and allosteric cavities across GPCR topological surfaces and allows users to select one of them (per run). Users are provided with three options to choose cavities in the Gcoupler: (1) one of the identified cavities from the Gcoupler, (2) providing key amino acids to receive cavity suggestions, or utilizing third-party tools for cavity identification and providing amino acid information. The synthesizer also performs in silico de novo synthesis of the active compounds within the target cavity, complementing the pharmacophores in the Cartesian space based on the target cavity. In summary, Synthesizer, the first module, generates and outputs the simplified molecular-input line-entry system (SMILES) and PDBQT files of the synthesized ligands along with cavity information in the form of grid coordinates, which is required as input by the next module.

Authenticator, the next module of Gcoupler, takes the PDBQT files and the cavity grid information generated using Synthesizer as input and further allows the segregation of these synthetic compounds based on their actual interaction at the molecular level. The Authenticator module leverages AutoDock Vina (1.2.3) python library for virtual screening (*Trott and Olson, 2010*). The binding energies produced by the Authenticator module are further used to segregate the synthetic compounds into High-Affinity Binders (HABs) and Low-Affinity Binders (LABs) based on the distribution while preserving the class balance for the downstream deep learning steps. Of note, the default value of binding energy for the distribution split is set to –7 kcal/mol; however, the module provides enough flexibility to the users to try and test different cutoffs (loose to stringent) and can visualize the change in distribution and statistically compare the resulting distributions within the Gcoupler workflow.

The Authenticator uses the Kolmogorov–Smirnov test (*Berger and Zhou, 2014*), Anderson–Darling test (*Engmann and Cousineau, 2011*), and Epps–Singleton test (*Goerg and Kaiser, 2009*) for hypothesis testing for the comparison of the distributions. The Authenticator module supports the graphical plots, such as overlapping density plots and Empirical Cumulative Distribution Function (ECDF), to visualize the distributions. In case either the Authenticator module fails to recognize the default cutoff as an optimal threshold for the distribution split or the user decides that the resulting distributions fail to obtain the statistical inference under any threshold of their choice, the module is also incorporated with the other options for providing negative datasets, such as decoys (Gcoupler inbuilt function using Chem module of RDKit Python package; *Landrum, 2023*) and user-supplied negative datasets.

The third module of Gcoupler, known as Generator, leverages the DeepChem (2.6.1) (*Ramsundar et al., 2019*) python library to build the Graph-Neural Networks supported classification models. The Generator takes the segregated HAB and LAB/decoys from the Authenticator module and uses

these synthetic compounds to build prediction models. By default, the Generator module supports four different graph-based models, i.e., Graph Convolution Model (GCM), Graph Convolution Network (GCN) (*Kipf and Welling, 2016*), Attentive FP (AFP) (*Xiong et al., 2020*), and Graph Attention Network (GAT) (*Veličković et al., 2018*), to generate classification models. Before the model-building step, the Generator module evaluates the class imbalance in the synthetic dataset and implements the upsampling techniques to counter it, if required. The Generator, by default, tests the preprocessed synthetic data on all four deep learning-based models under the default hyperparameters and returns models' performance parameter list, allowing users to select the best-performing model for the downstream refinement process. Users can either manually set up the hyperparameter tuning grid parameters or opt for the recommended grid range along with the fold count for *k*-fold cross-validation to tune the selected model around the best hyperparameters and evaluate the stability of the said optimized parameters. In the final step, the Generator builds the resultant model with the best-performing hyperparameters by utilizing the complete synthetic compound library (HAB + LAB/decoys) for training, allowing large-scale screening of the user query compounds by using SMILES information.

The final module of Gcoupler, the BioRanker module, provides a framework for post-data analysis. It processes prediction outputs from the Generator module to conduct functional activity-based subsetting and screening of a query compound library. Initially, it employs a stringent prediction probability threshold to select positively predicted query compounds. Users can also choose the inbuilt function with *G*-means algorithm and Youden's *J* statistics for more precise selection of query compounds to be processed. The geometric mean (*G*-mean) is the mathematical square root of the product of sensitivity and specificity, whereas Youden's *J* statistic (sensitivity + specificity − 1) is defined as the distance between the ROC curve and the chance line. The optimal threshold is the one that maximizes the J Statistics. These selected compounds are then projected onto various biological activity spaces, including Chemistry, Targets, Networks, Cells, and Clinics, through pairwise cosine similarity comparisons between their biological activity descriptor vectors and those of synthetic HABs. Subsequently, a modified PageRank algorithm is utilized for activity-specific ranking of the query compounds. Additionally, the module supports multi-activity-based ranking, enabling sequential screening based on a user-defined list of biological properties, thus enhancing the precision and relevance of compound selection.

## Runtime analysis

To showcase the speed and efficiency of our de novo coupled Deep Learning approach against conventional docking, we performed a comparative study involving analysis of the same query compound dataset against a single target of interest through both approaches. A system with 125 GB RAM, 16 Threads, and 8 CPU cores with no upgradation to the software or hardware within the duration of this analysis was used as the platform for this comparison. To compare the time complexity between Gcoupler and classical docking procedures using AutoDock (*Morris et al., 2009*), we selected human alpha-1A adrenergic receptors (ADRA1A) as our GPCR of interest. Since the experimentally elucidated structure of ADRA1A is unknown, we used AlphaFold (*Jumper et al., 2021*) predicted structure as our starting point. The ligand binding cavity at the extracellular site was determined using the LigBuilder V3.0 (*Yuan et al., 2020*) cavity function. Moreover, we obtained the ADRA1A ligand information from the ChEMBL database (Version 31) corresponding to the accession ID: CHEMBL229 (*Gaulton et al., 2012*). We obtained a total of 3684 bioactive compounds against ADRA1A; however, to ensure uniformity, we only selected those ligands (#933) that were annotated as 'single protein format experiments'. Files preparation for AutoDock-based docking includes conversion of ligand SMILES to MOL2 using OpenBabel (2.4.1) (*O'Boyle et al., 2011*), followed by conversion of MOL2 to PDBQT via MGLtools (1.5.6) (*Dallakyan, 2021*), and receptor PDB to PDBQT using OpenBabel (2.4.1) (*O'Boyle et al., 2011*). Noteworthily, the grid parameter calculations were manually performed, and docking was performed using AutoDock (4.2.6).

The Gcoupler workflow was performed using the steps above. To ensure uniformity of the target cavity between both approaches, the cavity information, along with the grid coordinate obtained during the initial steps of the AutoDock approach (*Morris et al., 2009*), was reused by the Synthesizer and the Authenticator module for generating de novo ligands and their molecular interaction-based classification, respectively. The Authenticator module selected an optimal binding energy cutoff

of –8 kcal/mol; however, we opted for decoys for the negative dataset generation due to severe class imbalance. Under default hyperparameters, AttentiveFP showed better overall performance scores among the four deep learning-based models tested. We used the AttentiveFP model with the following hyperparameters: {'num_layers': 2, 'num_timesteps': 2, 'graph_feat_size': 200, 'dropout': 0} after assessment with 10-fold cross-validation. The final model was generated using the complete HAB compound library as the positive and respective decoys dataset as a negative class for the model learning. Post-model generation, all 933 ligands were screened for their binding prediction probabilities.

## Sequence–structural–functional level analysis

All the available, non-redundant 66 human GPCR–Gα complexes were downloaded from the RCSB PDB database (https://www.rcsb.org/). The sequence level information for each of these GPCRs was obtained from the UniProt database (https://www.uniprot.org). Multiple Sequence Alignment (MSA) was performed to study the sequence level conservation, using the FASTA sequences of the GPCRs as input for the Multiple Sequence Comparison by Log-Expectation (MUSCLE) algorithm (*Edgar, 2004*). MUSCLE uses the PAM substitution matrix and sum-of-pairs (SP) score for alignment scoring. The MUSCLE alignment output in CLW format was then subsequently used to compute amino acid level conservation using the WebLogo tool (https://weblogo.berkeley.edu/logo.cgi). To identify conserved motifs native to the GPCR–Gα interface, residues at the GPCR–Gα interface for each GPCR–Gα complex were retrieved using EMBL-EBI developed PDBePISA tool and subsequently mapped on the WebLogos (*Battle, 2016*). To test the structural level conservation, the pairwise similarity of the PDBePISA provided GPCR–Gα interfaces (cavity) of all 66 GPCRs was achieved using the normalized root mean square deviation (RMSD), calculated using the PyMOL software (https://pymol.org/2/). The RMSD values were normalized by dividing the cavity RMSD by their respective whole protein RMSD. Finally, to analyze the functional level conservation of the GPCR-Gα interfaces, we used the Gcoupler's Synthesizer module to compute 50 synthetic ligands for each GPCR–Gα interface (cavity) and calculated their physicochemical properties (descriptors) using Mordred (*Moriwaki et al., 2018*). For each GPCR–Gα-protein interface, we computed a single vector representing the aggregated physicochemical property of all the synthesized compounds for the given GPCR cavity and measured their similarities using the Cosine similarity.

## Molecular dynamics simulation

The molecular dynamics simulation was performed using either GROMACS (*Abraham et al., 2015*) or AMBER (*Case et al., 2005*) software suite. For Ste2p, the experimentally elucidated structure with PDB ID: 7ad3 was downloaded from the RCSB PDB database (*Velazhahan et al., 2021*). Since the available structure was provided for the dimer, PDB preprocessing steps were used to isolate a single chain (chain B) that refers to the GPCR. Embedding in the lipid bilayer representing yeast plasma membrane composition was performed using the CHARM-GUI (*Jo et al., 2008*) and the simulations using GROMACS (*Abraham et al., 2015*). Notably, we performed six rounds of equilibration with the membrane system and nine rounds of production runs of 5 ns each. We finally selected the last frame of the simulation step from the production run for further analysis. Protein–ligand complexes were prepared for the molecular dynamics simulations using AMBER software (*Case et al., 2005*), as discussed above. Notably, we performed all analyses using the Ste2p monomers. The grid box center and several grids were chosen appropriately to restrict the conformational sampling within the specific binding site. By default, the software estimates docking energies for 250,000 conformations of the ligands within the binding sites and ranks them to find the top 10 binding modes. The best binding modes for the ligands in these two sites were analyzed. The subsequent molecular dynamics simulations were performed for the ligands in these two binding sites, and the input configuration is based on the aforementioned molecular docking study results. The charges for the ligands were computed using density functional theory calculations at the B3LYP/6–31+G* level of theory. We used a general amber force field (GAFF) to describe the molecular interactions (*Sprenger et al., 2015*). For protein, we employed the FF19SB force field (*Tian et al., 2020*), and for water, the TIP3P force field was used (*Wang and Yang, 2018*). The Ste2p–ligand complexes were embedded in about 23,000 water molecules. The minimization run, simulation in the constant volume ensemble, and simulation in the isothermal-isobaric ensemble were carried out. The time step for integrating the equation of

motion was 2 fs. Following the equilibration run, a time scale simulation was carried out for 100 ns (three replicates for IC4 and IC5) and 550 ns (one replicate for IC4 cavity only). All the Ste2p–ligand complex simulations were performed using Amber20 software (*Case et al., 2005*). Various structural properties, such as RMSD and RMSF, were computed and analyzed. The binding free energies using molecular mechanics-Generalizer Born surface area approach (MM/GBSA) were computed for 1000 configurations picked up from the last 10 ns trajectory. The binding free energies computed using the molecular mechanics-generalized Born surface area approach are the quantitative measure of the binding preference of ligands to a specific target binding site. The values are obtained from the time average of different contributions such as van der Waals, electrostatic, polar, and non-polar solvation energies. When compared to molecular docking approaches, these binding free energies are obtained as an average over multiple protein–ligand configurations from MD simulations, and so are more reliable and accurate. In addition, many of the molecular docking simulations do not account for conformational flexibility. Still, in the MM/GBSA calculations, the effects of protein-conformational flexibility, segmental motion, and cavity breathing are also accounted for, making results more reliable compared to molecular docking-based binding affinities.

## Molecular docking (AutoDock)

YMDB metabolites were downloaded from the YMDB database (Version 2.0) (*Jewison et al., 2012*). Of note, we took the metabolites structure file (SDF format) provided on the YMDB database to extract 2D information of the metabolites, and to gain missing metadata, we intersected these metabolites with the YMDB full database and obtained a total of 1264 metabolites. When we finally converted these compounds into 3D structures for docking or into graphs for testing data for the DeepChem-based model, a fraction of these compounds failed to convert, and we finally selected a unique set of 1198 metabolites. The file preparation for the docking protocol involved converting the YMDB SMILES (1D) to MOL2 (3D) format. This was achieved using the OpenBabel software (2.4.1) (*O'Boyle et al., 2011*). Further, the compounds from the MOL2 format were converted into dockable PDBQT format by following the MGLtools (1.5.6) ligand preparation procedure (*Dallakyan, 2021*). Of note, the sub-processing steps include detecting the ligand center, selecting single bonds as torsion/rotatable bonds, etc. As discussed above, the receptor PDBs were also converted into PDBQT format using the OpenBabel software (*O'Boyle et al., 2011*). AutoDock (4.2.6) (*Morris et al., 2009*) was used for molecular docking by implementing a Genetic algorithm with the following parameters (population size = 150, maximum number of evaluations = 25,000,000, and maximum number of generations = 27,000). Notably, 10 complexes were generated per YMDB compound. The docking results, that is, extracting the lowest binding energy from each docked protein–ligand complex, were analyzed using the MGLtools (1.5.6) (*Dallakyan, 2021*). Notably, the ΔG values were calculated on the pre-simulated stable docked structure. In the case of blind docking, the grid was made across the complete Ste2p structure and was docked using the aforementioned parameters with the YMDB metabolites.

## Functional enrichment analysis

Functional enrichment analysis of the differentially enriched metabolites or the metabolites of interest was performed using MetaboAnalyst (https://www.metaboanalyst.ca/) web server (*Chong et al., 2018*). For the YMDB compound library, the union of the lead compounds identified for each target cavity (IC4 and IC5) was used as input. In the case of the untargeted metabolomics data, the differentially enriched metabolites between the α-factor treated and untreated conditions were used as input. The hypergeometric test was used for hypothesis testing and providing subsequent p-values for each resultant enrichment pathway.

## Protein–protein docking

To determine the impact of metabolite binding on GPCR–Gα and GPCR–α-factor interaction, we performed Protein–Protein docking using the HADDOCK (High Ambiguity Driven protein–protein DOCKing) webserver (*van Zundert et al., 2016*). HADDOCK is an innovative information-guided, flexible docking methodology for biomolecular complex modeling, uniquely utilizing ambiguous interaction restraints (AIRs) derived from identified or predicted protein interfaces, along with

explicit distance restraints (e.g., MS cross-links) and diverse experimental data like NMR residual dipolar couplings, pseudo contact shifts, and cryo-EM maps, setting it apart from ab initio docking approaches. In the case of Ste2p-miniGpa1p or Ste2p–α-factor docking, the PDB chains of both proteins were extracted from the PDB complex (accession id: 7ad3) (*Velazhahan et al., 2021*). Ste2p in the docked complex with the respective metabolites was simulated via the AMBER software suite (*Case et al., 2005*). Post-simulation, the most stable conformation of protein-metabolite was taken, and the metabolite was manually removed to obtain a metabolite-influenced stable Ste2p structure. HADDOCK was subsequently used for the protein–protein docking between wild Ste2p or metabolite-influenced bound stable Ste2p against miniGpa1–protein or α-factor (extracted from PDB ID: 7ad3). The optimal representative complexes were obtained from the HADDOCK results (*van Zundert et al., 2016*) and were used as input in PRODIGY (PROtein binDIng enerGY prediction) (*Xue et al., 2016*) for $K_d$ and $\Delta G$ calculations. PRODIGY is a web service that provides support for the prediction of binding affinity in biological complexes. Notably, the $\Delta G$ values were calculated on the pre-simulated stable docked structures.

## Yeast strains

BY4741 (*MATa his3Δ1 leu2Δ0 met15Δ ura3Δ*) and *ste2Δ* (*MATa ste2Δ::kanMX his3Δ1 leu2Δ ura3Δ met15Δ*) strains of *Saccharomyces cerevisiae* were used in all the experiments. For the mating assay, BY4742 strain (*MATα his3Δ1 leu2Δ0 lys2Δ0 ura3Δ0*) was also used. All the knockouts were obtained from the Yeast Deletion Collection (*MATa xxx::kanMX his3Δ1 leu2Δ0 ura3Δ0 met15Δ0*). Unless mentioned, the yeast was grown at 30°C at 200 rpm in yeast extract–peptone–dextrose (YPD) (1% yeast extract, 2% peptone, and 2% dextrose). Agar (1.5%) was additionally added to YPD to prepare plates. Both the site-directed yeast mutants (T155D, L289K, and S75A) as well as reconstituted *STE2* transgenes (rtWT) were grown in yeast extract–peptone–galactose-raffinose (YPGR; 1% yeast extract, 2% peptone, and 2% galactose, 2% raffinose).

## Metabolomics

Wildtype (BY4741) and *ste2Δ* yeast strains were inoculated in the primary culture, followed by a secondary culture, in YPD medium at 30°C at 150 rpm for 16 hr each. 1.5 ml of the secondary cultures were aliquoted into a 96-well deep well plate to ensure an equal number of cells for both strains. α-factor (T6901, Sigma-Aldrich) was added to the wells to attain final concentrations of 10, 20, 30, 40, and 50 μM, with eight biological replicates each. DMSO (28580, SRL) (volume equivalent to the 30 μM α-factor sample) was added to the solvent control. DMSO or α-factor was not added to untreated (WT) and *ste2Δ* conditions. The plate was covered with a breathe-easy membrane (Z380059, Sigma-Aldrich) and incubated at 30°C at 150 rpm for 4 hr. After incubation, an aliquot of 50 μl was used to perform propidium iodide (PI) fluorometric assay as described in the later section.

Post PI-based cell viability assay, four biological replicates were created by pooling biological replicates. These were transferred to 1.5 ml microcentrifuge tubes and pelleted down at 6000 rpm for 5 min at room temperature (RT). The cell pellet was treated with zymolyase enzyme (L2524, Sigma-Aldrich) at a final concentration of 40 units/mL in 1X PBS and incubated at 30°C for 1 hr. Following this, the cells were washed with 1X PBS, and the cell pellet was stored at –80°C before metabolomics.

Metabolomics was performed on 1290 Infinity HPLC coupled with 6545 QTOF (Agilent Technologies, Santa Clara, CA, USA) equipped with dual ESI Source. The internal controls used in metabolomics include Jasmonic acid, Gibberellic acid, zeatin, Tryptophan 15N, Estrone D4, Arginine 15N, Thiamine D4, 15N5-8-OH-2-dG, 13C12 dityrosine, and 8-PGF2-D4. The protein precipitation method was used for the sample preparation. Briefly, 10 μl of a cocktail containing commercial stable isotopes internal standards were added, followed by the addition of chilled 750 μl methanol: water (4:1) and probe sonicated. Then, 450 μl of chilled chloroform was added and vortexed for 5 min, followed by adding 150 μl of chilled water, vortexed, and kept the sample at –20°C for 30 min. Next, samples were centrifuged at 5000 rpm at 4°C for 10 min to separate the methanol and chloroform layers. Both layers were collected and dried at 30°C using a vacuum concentrator and reconstituted in 100 μl methanol: water (1:1). Quality Control (QC) samples were injected between every five samples to check the drift in the instrument and to evaluate the signal repeatability. Pearson's

rank order showed that QCs ($n$ = 6) were strongly correlated ($r > 0.9$) with each other, suggesting variations were insignificant.

Data analysis of the metabolomics data includes peak normalization. Furthermore, the normalized peak intensities of all the samples' replicates were merged, and the metabolites with a constant value or with more than 50% missing values were omitted. The missing values were imputed using the feature-based $k$ nearest neighbors (kNN) via Metaboanalyst webserver. The data were filtered based on the interquartile range (IQR). To find the list of differentially enriched metabolites (DEMs), $\log_2$ fold change ($\log_2$FC) values were calculated for each sample with respect to the solvent control, along with its statistical significance using Student's $t$-test. The metabolites with $|\log_2$FC $\geq 1|$ and p-value $<0.05$ were considered DEMs. Pathway Over Representation Analysis (ORA) was performed using MetaboAnalyst (*Chong et al., 2018*). Note: ORA is a computational method used in metabolomics research to gain insights into the biological significance of a set of metabolites that exhibit changes under different experimental conditions or in response to a perturbation or those belonging to a particular group. This analysis is conducted using statistical tests such as hypergeometric tests or Fisher's exact tests to determine if the observed number of metabolites in a given pathway is significantly higher than expected based on the background distribution of metabolites.

## Genetic screening

Fifty-three knockout strains from the Yeast Deletion Collection, along with WT and *ste2Δ* strains, were grown in primary culture, followed by secondary culture, in YPD medium at 30°C at 150 rpm for 16 hr each. Ensuring an equal number of cells in all the strains, 5 µl of secondary culture was inoculated into 96-well plates containing 145 µl of YPD and grown for 16 hr at 30°C at 200 rpm with eight biological replicates. Afterward, α-factor was added to half the replicates at a final concentration of 30 µM. An equal volume of DMSO was added to the other half to serve as solvent control. The plates were incubated for another 4 hr. After incubation, a 50 µl aliquot was mixed with 50 µl of staining solution (PI (11195, SRL) in 1x PBS, freshly prepared) to attain a final PI concentration of 5 µg/ml in 96-well black plates. The plates were incubated in the dark for 15 min. Heat-killed (HK) cells were used as a positive control. The fluorescence was measured using Biotek Synergy HTX multi-mode reader at excitation and emission wavelengths of 530/25 nm and 590/25 nm, respectively. Another 50 µl aliquot from the original plates was used to measure $OD_{600}$ after 1:2 dilution with 1X PBS. The fluorescence values were first normalized for data analysis with the blank normalized $OD_{600}$ values of the respective well. These normalized fluorescence values were subjected to two more rounds of normalization, first with unstained cells followed by HK cells. Following this, percentage fold change was calculated for the treated group with respect to the untreated group. The p-value was computed using a one-sample Student's $t$-test.

## Pre-loading of yeast cells with metabolite

Yeast cells were cultured in YPD medium at 30°C, 200 rpm for 16 hr in primary and secondary cultures. Equal cell densities (5 µl) from secondary cultures were inoculated into 96-well plates containing 145 µl YPD with metabolites: coenzyme Q6 (CoQ6, 900150O, Avanti Polar Lipids), zymosterol (ZST, 700068P, Avanti Polar Lipids), and lanosterol (LST, L5768, Sigma-Aldrich) at 0.1, 1, and 10 µM concentrations. Plates were incubated for 24 hr at 30°C, 200 rpm, with multiple biological replicates. Ethanol-treated wells served as solvent controls.

Site-directed Ste2 mutants were grown in YPD for primary and secondary cultures, but metabolite pre-loading was performed in YPGR instead of YPD to induce Ste2 expression.

## Cell viability assay using growth kinetics

Wildtype and *ste2Δ* yeast strains were grown and pre-loaded with ZST, LST, and CoQ6 at concentrations of 0.1, 1, and 10 µM. After pre-loading, 5 µl cultures from this plate were used as an inoculum for another 96-well plate containing 145 µl of YPD along with respective metabolites (CoQ6, ZST, and LST) at concentrations of 0.1, 1, and 10 µM each in the presence of 10 µM α-factor (T6901, Sigma-Aldrich) in the treated group and DMSO in the untreated group with multiple biological replicates. The plates were sealed with breathe-easy membranes to prevent evaporation. The growth kinetics was performed in a Biotek Synergy HTX multi-mode reader for 16.5 hr at 30°C under continuous orbital shaking at 559 CPM. Absorbance was measured at 600 nm every 30 min.

The area under the curve (AUC) was calculated for each sample using the bayestestR package in R with default parameters and normalized using the $OD_{600}$ value at time $t = 0$.

## Cell viability assay using PI

Wild-type and *ste2Δ* yeast strains were grown and pre-loaded with CoQ6, ZST, and LST at concentrations of 0.1, 1, and 10 µM for 24 hr. Following pre-loading, the α-factor was added to half the wells at a final concentration of 30 µM. An equal volume of DMSO was added to the other half to serve as solvent control. The plates were incubated for another 4 hr. After incubation, a 50 µl aliquot was mixed with 50 µl of staining solution (PI in 1X PBS, freshly prepared) to attain a final PI concentration of 5 µg/ml in 96 well-black plates. The plates were incubated in the dark for 15 min. HK cells were used as a positive control. The fluorescence was measured using Biotek Synergy HTX multi-mode reader at excitation and emission wavelengths of 530/25 nm and 590/25 nm, respectively. Another 50 µl aliquot from the original plate was used to measure $OD_{600}$ after 1:2 dilution with 1X PBS.

The fluorescence values were first normalized for data analysis with the blank normalized $OD_{600}$ values of the respective well. Cellular background fluorescence values of the unstained cells were subtracted from these normalized fluorescence values. Post this, the fold change between the untreated solvent control and the α-factor treated samples was computed by dividing the latter with the former. Mann–Whitney *U*-test was used to measure the statistical significance.

The assays for site-directed Ste2 mutants and differentially upregulated gene knockouts were performed and analyzed in the same way with the following modifications: the site-directed Ste2 mutants were grown in YPGR instead of YPD, and the knockouts were grown in the presence of 1 µM LST for pre-treatment before α-factor treatment.

## Cell viability assay using FUN1 staining

Wildtype and *ste2Δ* yeast strains were pre-loaded with metabolites at 1 µM (ZST and LST) and 10 µM (CoQ6) concentrations and grown for 24 hr with nine biological replicates. An equal volume of ethanol was added to the control group to negate the effects of the solvent. Following this, α-factor was added to a third of the wells at a final concentration of 30 µM, acetic acid (85801, SRL) to another third at a final concentration of 199 mM, and DMSO to the remaining third to serve as solvent control. The plates were incubated for another 4 hr. Post-incubation, the cells were pelleted down and washed with 1X PBS to remove any media traces at 6000 rpm for 5 min at RT. The staining solution was freshly prepared in 1X GH solution (10 mM HEPES buffer, pH 7.2, 2% glucose) with FUN 1 Cell Stain (F7030, Invitrogen) at a final concentration of 5 µM. The samples were incubated in the dark for 30 min, centrifuged at 6000 rpm for 5 min, and the pellet resuspended in 25 µl of 1x GH solution. 5 µl of the sample was used for fluorescence microscopy (Nikon Eclipse Ci-L Fluorescence Microscope with Nikon DS-Qi2 camera). FITC and TRITC filters were used for measuring green and red fluorescence, respectively, with an exposure time of 1 s each. For each sample, the number of cells with red and green fluorescence, indicating cellular viability and vitality in each image, was counted using ImageJ and normalized with the total cell counts. The assay for site-directed Ste2 mutants was performed and analyzed similarly, with the only difference being that the yeast cells were pre-loaded with metabolites in YPGR instead of YPD.

## Mating assay

BY4741 wild-type yeast strains were pre-loaded with metabolites at 1 µM (ZST and LST) and 10 µM (CoQ6) concentrations and grown for 24 hr with biological triplicates. After 24 hr, the cells were harvested, washed (6000 rpm, 5 min, RT) to remove metabolite traces, and resuspended in fresh YPD. Equalized volumes of treated MATa and untreated MATα cells (100 µl each) were mixed in 96-well plates, sealed with a breathable membrane, and incubated for 4 hr at 30°C, 200 rpm to allow mating. Post-mating, cells were washed with distilled water (6000 rpm, 5 min, RT), resuspended in 200 µl distilled water, and serially diluted (1:10). Diluted cultures (5 µl) were spotted on SC-Met (His, Leu, Ade, Ura, Lys), SC-Lys (His, Leu, Ade, Ura, Met), and SC-Met-Lys (His, Leu, Ade, Ura) agar plates (composition: 2% glucose, 2% agar, 20 mg/l histidine, 120 mg/l leucine, 60 mg/l lysine, 20 mg/l methionine, 20 mg/l uracil, and 20 mg/l adenine) in technical triplicates. Plates were incubated at

30°C for 2 days. After incubation, the colonies were counted in ImageJ in the $10^4$ dilutions, which had the best resolution of colonies. Mating efficiency was calculated using the following formula:

$$Mating\ Efficiency = \frac{colonies\ in\ DD\ \times\ Dilution\ Factor}{colonies\ in\ SD\ \times\ Dilution\ Factor} \times 100.$$

Here, SD is single dropout media (SC-Met for MATα, and SC-Lys for MATa), and DD is double dropout (SC-Met-Lys).

## Phospho-MAPK activity-based western blot

The wild-type BY4741 strain was grown in primary culture, followed by secondary culture, in YPD medium at 30°C at 200 rpm for 16 hr each. 30 μl of secondary culture was inoculated into 1.5 ml of YPD with respective metabolites at concentrations of 1 μM (ZST and LST) and 10 μM (CoQ6) and grown for 24 hr at 30°C at 200 rpm with 6 biological replicates, with ethanol as solvent control. After incubation, the cells were treated with 3 μM of α-factor for 5 min with DMSO as solvent control. Post-incubation, the cells were harvested at 6000 rpm for 5 min at RT. The pellet was resuspended in 400 μl of chilled Lysis buffer (9.5 ml 1X PBS, 200 μl of Protease Inhibitor Cocktail, 200 μl of PMSF (20 mg/ml), 100 μl of 10% Triton X-100) with an equal volume of glass beads. The cells were homogenized for five cycles of 30 s at 3500 rpm, followed by 1-min incubation in ice each. The beads were allowed to settle down with a short spin, following which the lysate was transferred to a fresh vial. The lysate was centrifuged at 2000 rpm for 2 min at 4°C. Supernatants were then transferred to fresh vials, and 5 μl were used in a BCA Protein Assay (Thermo Scientific #23225) carried out according to the manufacturer's instructions. Absorbance values were compared against bovine serum albumin standards. Lysates were normalized with lysis buffer to 470 μg/μl, mixed 1:4 with 4X Sample Loading buffer (2 ml 1 M Tris pH 6.8, 0.8 g SDS, 4 ml 100% glycerol, 0.4 ml β-mercaptoethanol, 1 ml 0.5 M EDTA, 8 mg bromophenol blue), and heated at 99°C for 10 min.

Ten milligrams of protein sample were loaded onto 12% SDS-PAGE gel and resolved in SDS electrophoresis buffer at 40 mA. The resolving gel was then transferred to methanol-activated PVDF membranes at 300 mA for 120 min in transfer buffer (3 g Tris, 14.4 g glycine, 200 methanol, 800 ml water) at 4°C. The membranes were placed in a blocking buffer (5% (wt/vol) BSA for p-Fus3 or 5% (wt/vol) skim milk in 1X TBST for rest) for 2 hr at room temperature, washed 3 × 5 min with 1X TBST, and then probed with antibodies to phospho-p44/42 (p-Fus3; Cell Signaling Technology Cat# 4370, RRID:AB_2315112, 1:2000), Fus3 (Cell Signaling Technology Cat# 9102, RRID:AB_330744, 1:1000), or 3-Phosphoglycerate Kinase (Pgk1; Thermo Fisher Scientific Cat# 459250, RRID:AB_2532235, 1:5000) as a loading control in 5% BSA for 2 hr at room temperature with shaking. Blots were washed 3 × 5 min with 1X TBST and then incubated with horseradish peroxidase-conjugated goat anti-rabbit (Cell Signaling Technology Cat# 7074, RRID:AB_2099233, 1:5000), or horse anti-mouse (Cell Signaling Technology Cat# 7076, RRID:AB_330924, 1:5000) secondary antibodies in 1X TBST for 2 hr at room temperature. Blots were washed 3 × 5 min with 1X TBST and imaged on a UVTech imaging system after incubation with Immobilon Forte Western HRP Substrate (Millipore #WBLUF0500). Phospho-Fus3 antibody was removed by treatment with mild Stripping Buffer (15 g glycine, 1 g SDS, 10 ml Tween20, pH 2.2, make up to 1 l with water) for 10 min at RT with shaking, discarding the buffer and reincubating for 10 min in fresh stripping buffer, then rinsed thoroughly with 1X TBST 5 × 5 min before blocking and re-probing with Fus3 antibody. Blots were stripped once again and re-probed for Pgk1 as a loading control.

ImageJ was used to perform the densitometric analysis of the blots. The raw densitometric values so obtained for p-Fus3 and Fus3 were normalized with the loading controls, and the p-Fus3/Fus3 ratio was calculated for each condition. For mutants, this ratio for the samples stimulated with α-factor was further normalized with their corresponding unstimulated solvent controls.

## Transgenic reporter assay

The pRS306-P$_{FUS1}$-eGFP-CYC1 plasmid was constructed by subcloning the eGFP gene under the control of the FUS1 promoter into pRS306 using Gibson assembly. The resulting plasmid was integrated at the *URA3* locus of the BY4741 strain, generating the genotype *MATa his3Δ1 leu2Δ0 met15Δ0 ura3::pRS306-P$_{FUS1}$-eGFP-CYC1*.

**Appendix 1—table 1.** The list of primers used in PCR amplification of indicated genes and plasmids for Gibson Assembly.

| Primer name | Sequence (5′–3′) |
| --- | --- |
| pFUS1 Forward | GTAAAACGACGGCCAGTGAGCTCAATCCTTCAATTTTTCTGGCAACTTTTCTC |
| pFUS1 Reverse | GCGTGACATAACTAATTACATGACTCGAGTTACTTGTACAGCTCGTCCATGCCG |
| eGFP Forward | CCATCAAGTTTCTGAAAATCAAAGGATCCATGAGTAAGGGCGAGGAGCTGTTCACCG |
| eGFP Reverse | GCGTGACATAACTAATTACATGACTCGAGTTACTTGTACAGCTCGTCCATGCCG |
| pRS306 Forward | CTCGAGTCATGTAATTAGTTATGTCACG |
| pRS306 Reverse | GAGCTCACTGGCCGTCGTTTTAC |

The transformed strain was cultured in SC-Ura medium (2% glucose, 200 mg/l histidine, 300 mg/l leucine, 400 mg/l methionine, 200 mg/l tryptophan, and 100 mg/l adenine) at 30°C, 200 rpm for 16 hr in both primary and secondary cultures. Equalized cell densities (5 µl) were inoculated into 96-well plates containing 145 µl SC-Ura medium with metabolites (ZST, LST: 1 µM; CoQ6: 10 µM) or ethanol (control) and incubated for 24 hr under identical conditions (three biological replicates). Following 24 hr, α-factor or DMSO (control) was added at a final concentration of 30 µM and incubated for a total of 1.5 hr. Toward the end of the incubation, PI (5 µg/ml) was added to co-stain for 15 min in the dark. Samples (10 µl) were analyzed using Nikon A1R Confocal/TIRF/FLIM microscopy. GFP (488 nm) and mCherry (561 nm) filters were used for measuring green and red fluorescence, respectively. The images were captured at a scan size of 1024 pixels, scan speed of ¼ frame/s, and line average at a count of 4, and the data analysis was performed using ImageJ (*Schindelin et al., 2012*). Dead cells were excluded based on mCherry fluorescence intensity above 2500 intensity units, and cells with an area below 3 pixels were removed to reduce noise in the data. Corrected Total Cell Fluorescence (CTCF) was calculated from 500 randomly selected cells per condition.

$$CTCF = IntDens - \left( Area\,of\,the\,selected\,cell\, * MF \right).$$

Here, IntDen is the Integrated Density of the selected cell, and MF is the Mean fluorescence of the background readings.

## Site-directed mutagenesis

The gene encoding wild-type *STE2* was PCR amplified from the genome of *Saccharomyces cerevisiae* and further cloned into plasmid pRS304 under galactose-inducible *GAL1* promoter and CYC1 terminator to generate plasmid pRS304-P$_{GAL1}$-STE2-CYC1 using Gibson assembly (*Gibson et al., 2010*; *Gibson et al., 2009*). The mutants were generated by PCR amplifying the gene with primers consisting of respective mutations and cloned into plasmid pRS304 under *GAL1* promoter to generate pRS304-P$_{GAL1}$-STE2 S75A-CYC1, pRS304-P$_{GAL1}$-STE2 T155D-CYC1, and pRS304-P$_{GAL1}$-STE2 L289K-CYC1. Wild-type (rtWT) and mutant *STE2* were integrated by digesting the pRS304 vector with restriction enzyme BstXI to generate linearized plasmid and transformed into *Saccharomyces cerevisiae* BY4741 *ste2Δ* strain. The strains were confirmed for integration by isolating the genome and PCR amplification using locus-specific primers (*Appendix 1—table 2*). The following mutants were generated in this work:

a. *MATa his3Δ1 leu2Δ0 met15Δ0 ura3Δ0 ste2Δ::kanMX TRP1::P$_{GAL1}$-STE2-CYC1*
b. *MATa his3Δ1 leu2Δ0 met15Δ0 ura3Δ0 ste2Δ::kanMX TRP1::P$_{GAL1}$-STE2 S75A-CYC1*
c. *MATa his3Δ1 leu2Δ0 met15Δ0 ura3Δ0 ste2Δ::kanMX TRP1::P$_{GAL1}$-STE2 T155D-CYC1*
d. *MATa his3Δ1 leu2Δ0 met15Δ0 ura3Δ0 ste2Δ::kanMX TRP1::P$_{GAL1}$-STE2 L289K-CYC1*

**Appendix 1—table 2.** The list of primers used in cloning and site-directed mutagenesis.

| Primer name | Sequence (5′–3′) |
| --- | --- |
| STE2 FWD overlap | CTTTAACGTCAAGGAGGGATCCATGTCTGATGCGGCTCCTTCATTG |

*Appendix 1—table 2 Continued on next page*

*Appendix 1—table 2 Continued*

| Primer name | Sequence (5′–3′) |
| --- | --- |
| SDM1 STE2 FWD | GTCATGTGGATGACATCGAGAGCTAGAAAAACGCCGATTT |
| SDM1 STE2 REV | AAATCGGCGTTTTTCTAGCTCTCGATGTCATCCACATGAC |
| SDM4 STE2 FWD | TTTCAGATAAAAGTTATTTTCGACGGCGACAACTTCAAAAGGATA |
| SDM4 STE2 REV | TATCCTTTTGAAGTTGTCGCCGTCGAAAATAACTTTTATCTGAAA |
| SDM5 STE2 FWD | ACATTACTTGCTGTATTGTCTAAACCATTATCATCAATGTGGGCC |
| SDM5 STE2 REV | GGCCCACATTGATGATAATGGTTTAGACAATACAGCAAGTAATGT |
| STE2 REV overlap | CATAACTAATTACATGACTCGAGTCATAAATTATTATTATCTTCAGTCCAGAACTTTCTG |
| CYC.F | ATAATAATTTATGACTCGAGTCATGTAATTAGTTATGTCACGCTTAC |
| GAL1 REV BAMHI | GCCGCATCAGACATGGATCCCTCCTTGACGTTAAAGTATAGAGG |

## Cellular morphometric assay (shmoo formation)

The reconstituted *STE2* wild-type (rtWT) and mutants (T155D, L289K, and S75A) were grown in primary culture, followed by secondary culture, in YPGR medium at 30°C at 200 rpm for 16 hr each. Ensuring an equal number of cells for all, 5 µl of secondary culture was inoculated into a 96-well plate containing 145 µl of YPD in the presence or absence of respective metabolites at desired concentrations (T155D – 1 µM LST, L289K – 1 µM ZST, and S75A – 10 µM CoQ6) and grown for 24 hr at 30°C at 200 rpm with four biological replicates. An equal volume of ethanol was added to the control group to negate the effects of the solvent. After 24 hr, α-factor (T6901, Sigma-Aldrich) was added at a final concentration of 10 µM, with DMSO as solvent control, and incubated for 1 hr. Post-incubation, 5 µl of the sample was used for phase contrast microscopy (Nikon Eclipse Ci-L Fluorescence Microscope with Nikon DS-Qi2 camera). For each sample, the number of cells with shmoos was counted using ImageJ and normalized with the total cell counts.

## RNA-sequencing

Yeast cells (BY4741) were grown at 30°C at 200 rpm in YPD medium in biological duplicates in the presence or absence of lanosterol (LST) at a final concentration of 1 µM, with ethanol as control. Post-metabolite loading, the cells were either treated or untreated with the α-factor at a final concentration of 30 µM for 2 hr at desired growth conditions. Following these steps, the cells were harvested, and RNA was isolated using the protocol described in *Mittal et al., 2022*. Following sequencing, we conducted a quality control check with MultiFastQ. The paired-end sequencing reads were aligned to the yeast reference genome (ENSEMBL; R64-1-1; GCA_000146045.2) via the align function of the Rsubread package (v.2.6.4), generating aligned BAM files. Before mapping, read trimming was performed at both ends. An expression matrix was produced with the featureCounts function to contain raw read counts for each sample. The read count matrix was normalized using TMM (trimmed mean of M values) normalization (*Supplementary file 8*), and the differential gene expression analysis was performed using NOISeq (v2.38.0) (*Tarazona et al., 2015*; *Supplementary files 9–11*). The functional gene ontology analysis was performed using the Gene Ontology Term Finder tool (Version 0.86) of the *Saccharomyces* Genome Database (https://www.yeastgenome.org). The normalized read count matrix and the raw FASTQ files are available at the Zenodo repository.

## Cardiomyocyte hypertrophy models

Human cardiomyocyte cell line AC16 was cultured in DMEM-F12 (Thermo Scientific) with 12.5% fetal bovine serum (FBS; Thermo Scientific) at 37°C in the presence of 5% $CO_2$. AC16 cells were seeded in a 24-well cell culture plate for cell size measurements. After 24 hr, cells were treated with metabolites CoQ6, ZST, LST, FST (F5379, Sigma-Aldrich), and CoQ10 (15039, SRL) at the final concentration of 2.5 µM in the presence of 1% FBS and incubated overnight. The next day, the medium was changed. Fresh metabolites were added at 2.5 µM final concentration and isoproterenol treatment at 25 µM concentration and incubated for 48 hr in the presence of 1% FBS. Next, cells were washed with 1X

PBS, fixed with 4% paraformaldehyde, and stained with wheat germ agglutinin (Thermo Scientific) and DAPI. Images were taken by Leica DMI 6000 B (Leica Microsystems, USA) at 20X objective. Cell area was calculated by using ImageJ software.

Neonatal rat cardiomyocytes were isolated from 1- to 3-day-old SD rat pups using Collagenase Type II. Briefly, pups were sacrificed, hearts were explanted, washed in 1X PBS, and chopped into small pieces. Chopped heart pieces were digested with Collagenase Type II at 1 mg/ml (Thermo Scientific, USA) at 37°C. Digested cells are centrifuged, resuspended in a DMEM medium, and pre-plated for 90 min to remove fibroblasts. Cardiomyocytes containing supernatant were then collected and seeded in a gelatin-coated 24-well cell culture plate. After 48 hr, cells were washed with 1X PBS and incubated with metabolites at a concentration of 2.5 µM overnight in the presence of 1% FBS. The next day, the medium was changed, and cells were treated with metabolites (2.5 µM) and isoproterenol (10 µM) in the presence of 1% FBS. After 72 hr, cells were fixed and stained with alpha-sarcomeric actinin (#MA1-22863 Thermo Scientific, USA) and DAPI. Images were taken by Leica DMI 6000 B (Leica Microsystems, USA) at 20X objective. Cell area was calculated by using ImageJ software.

AC16 cells were provided by Dr. Regalla Kumar Swami, CSIR-CCMB, and primary cardiomyocytes were isolated from wild-type Sprague-Dawley (SD) rat pups (1–3 days old) male/female.

