## [Editor Report · eLife Assessment]

The authors present a set of wrappers around previously developed software and machine-learning toolkits, and demonstrate their use in identifying endogenous sterols binding to a GPCR. The resulting pipeline is potentially **useful** for molecular pharmacology researchers due to its accessibility and ease of use. However, the evidence supporting the GPCR-related findings remains **incomplete**, as the machine-learning model shows indications of overfitting, and no direct ligand-binding assays are provided for validation.

---

## [Referee Report · Reviewer #1 (Public review)]

This is a re-review following an author revision. I will go point-by-point in response to my original critiques and the authors' responses. I appreciate the authors taking the time to thoughtfully respond to the reviewer critiques.

Query 1. Based on the authors' description of their contribution to the algorithm design, it sounds like a hyperparameter search wrapped around existing software tools. I think that the use of their own language to describe these modules is confusing to potential users as well as unintentionally hides the contributions of the original LigBuilder developers. The authors should just explain the protocol plainly using language that refers specifically to the established software tools. Whether they use LigBuilder or something else, at the end of the day the description is a protocol for a specific use of an existing software rather than the creation of a new toolkit.

Query 2. I see. Correct me if I am mistaken, but it seems as though the authors are proposing using the Authenticator to identify the best distributions of compounds based on an in silico oracle (in this case, Vina score), and train to discriminate them. This is similar to training QSAR models to predict docking scores, such as in the manuscript I shared during the first round of review. In principle, one could perform this in successive rounds to create molecules that are increasingly composed of features that yield higher docking scores. This is an established idea that the authors demonstrate in a narrow context, but it also raises concern that one is just enriching for compounds with e.g., an abundance of hydrogen bond donors and acceptors. Regarding points (4) and (5), it is unclear to me how the authors perform train/test splits on unlabeled data with supervised machine learning approaches in this setting. This seems akin to a Y-scramble sanity check. Finally, regarding the discussion on the use of experimental data or FEP calculations for the determination of HABs and LABs, I appreciate the authors' point; however, the concern here is that in the absence of any true oracle the models will just learn to identify and/or generate compounds that exploit limitations of docking scores. Again, please correct me if I am mistaken. It is unclear to me how this advances previous literature in CADD outside of the specific context of incorporating some ideas into a GPCR-Gprotein framework.

Query 3. The authors mention that the hyperparameters for the ML models are just the package defaults in the absence of specification by the user. I would be helpful to know specifically what the the hyperparameters were for the benchmarks in this study; however, I think a deeper concern is still that these models are almost certainly far overparameterized given the limited training data used for the models. It is unclear why the authors did not just build a random forest classifier to discriminate their HABs and LABs using ligand- or protein-ligand interaction fingerprints or related ideas.

Query 4. It is good, and expected, that increasing the fraction of the training set size in a random split validation all the way to 100% would allow the model to perfectly discriminate HABs and LABs. This does not demonstrate that the model has significant enrichment in prospective screening, particularly compared to simpler methods. The concern remains that these models are overparameterized and insufficiently validated. The authors did not perform any scaffold splits or other out-of-distribution analysis.

Query 5. The authors contend that Gcoupler uniquely enables training models when data is scarce and ultra-large screening libraries are unavailable. Today, it is rather straightforward to dock a minimum of thousands of compounds. Using tools such as QuickVina2-GPU (https://pubs.acs.org/doi/10.1021/acs.jcim.2c01504), it is possible to quite readily dock millions in a day with a single GPU and obtain the AutoDock Vina score. GPU-acclerated Vina has been combined with cavity detection tools likely multiple times, including here (https://arxiv.org/abs/2506.20043). There are multiple cavity detection tools, including the ones the authors use in their protocol.

Query 6. The authors contend that the simulations are converged, but they elected not to demonstrate stability in the predicting MM/GBSA binding energies with block averaging across the trajectory. This could have been done through the existing trajectories without additional simulation.

---

## [Referee Report · Reviewer #1 (Public review)]

This is a re-review following an author revision. I will go point-by-point in response to my original critiques and the authors' responses. I appreciate the authors taking the time to thoughtfully respond to the reviewer critiques.

Query 1. Based on the authors' description of their contribution to the algorithm design, it sounds like a hyperparameter search wrapped around existing software tools. I think that the use of their own language to describe these modules is confusing to potential users as well as unintentionally hides the contributions of the original LigBuilder developers. The authors should just explain the protocol plainly using language that refers specifically to the established software tools. Whether they use LigBuilder or something else, at the end of the day the description is a protocol for a specific use of an existing software rather than the creation of a new toolkit.

Query 2. I see. Correct me if I am mistaken, but it seems as though the authors are proposing using the Authenticator to identify the best distributions of compounds based on an in silico oracle (in this case, Vina score), and train to discriminate them. This is similar to training QSAR models to predict docking scores, such as in the manuscript I shared during the first round of review. In principle, one could perform this in successive rounds to create molecules that are increasingly composed of features that yield higher docking scores. This is an established idea that the authors demonstrate in a narrow context, but it also raises concern that one is just enriching for compounds with e.g., an abundance of hydrogen bond donors and acceptors. Regarding points (4) and (5), it is unclear to me how the authors perform train/test splits on unlabeled data with supervised machine learning approaches in this setting. This seems akin to a Y-scramble sanity check. Finally, regarding the discussion on the use of experimental data or FEP calculations for the determination of HABs and LABs, I appreciate the authors' point; however, the concern here is that in the absence of any true oracle the models will just learn to identify and/or generate compounds that exploit limitations of docking scores. Again, please correct me if I am mistaken. It is unclear to me how this advances previous literature in CADD outside of the specific context of incorporating some ideas into a GPCR-Gprotein framework.

Query 3. The authors mention that the hyperparameters for the ML models are just the package defaults in the absence of specification by the user. I would be helpful to know specifically what the the hyperparameters were for the benchmarks in this study; however, I think a deeper concern is still that these models are almost certainly far overparameterized given the limited training data used for the models. It is unclear why the authors did not just build a random forest classifier to discriminate their HABs and LABs using ligand- or protein-ligand interaction fingerprints or related ideas.

Query 4. It is good, and expected, that increasing the fraction of the training set size in a random split validation all the way to 100% would allow the model to perfectly discriminate HABs and LABs. This does not demonstrate that the model has significant enrichment in prospective screening, particularly compared to simpler methods. The concern remains that these models are overparameterized and insufficiently validated. The authors did not perform any scaffold splits or other out-of-distribution analysis.

Query 5. The authors contend that Gcoupler uniquely enables training models when data is scarce and ultra-large screening libraries are unavailable. Today, it is rather straightforward to dock a minimum of thousands of compounds. Using tools such as QuickVina2-GPU (https://pubs.acs.org/doi/10.1021/acs.jcim.2c01504), it is possible to quite readily dock millions in a day with a single GPU and obtain the AutoDock Vina score. GPU-acclerated Vina has been combined with cavity detection tools likely multiple times, including here (https://arxiv.org/abs/2506.20043). There are multiple cavity detection tools, including the ones the authors use in their protocol.

Query 6. The authors contend that the simulations are converged, but they elected not to demonstrate stability in the predicting MM/GBSA binding energies with block averaging across the trajectory. This could have been done through the existing trajectories without additional simulation.

---

## [Author Response]

The following is the authors’ response to the original reviews

**Public Reviews:**

**Reviewer #1 (Public review):**
SummaryQuery: In this manuscript, the authors introduce Gcoupler, a Python-based computational pipeline designed to identify endogenous intracellular metabolites that function as allosteric modulators at the G protein-coupled receptor (GPCR) - Gα protein interface. Gcoupler is comprised of four modules:I. Synthesizer - identifies protein cavities and generates synthetic ligands using LigBuilder3II. Authenticator - classifies ligands into high-affinity binders (HABs) and low-affinity binders (LABs) based on AutoDock Vina binding energiesIII. Generator - trains graph neural network (GNN) models (GCM, GCN, AFP, GAT) to predict binding affinity using synthetic ligandsIV. BioRanker - prioritizes ligands based on statistical and bioactivity dataThe authors apply Gcoupler to study the Ste2p-Gpa1p interface in yeast, identifying sterols such as zymosterol (ZST) and lanosterol (LST) as modulators of GPCR signaling. Our review will focus on the computational aspects of the work. Overall, we found the Gcoupler approach interesting and potentially valuable, but we have several concerns with the methods and validation that need to be addressed prior to publication/dissemination.

We express our gratitude to Reviewer #1 for their concise summary and commendation of our work. We sincerely apologize for the lack of sufficient detail in summarizing the underlying methods employed in Gcoupler, as well as its subsequent experimental validations using yeast, human cell lines, and primary rat cardiomyocyte-based assays.

We wish to state that substantial improvements have been made in the revised manuscript, every section has been elaborated upon to enhance clarity. Please refer to the point-by-point response below and the revised manuscript.

Query: (1) The exact algorithmic advancement of the Synthesizer beyond being some type of application wrapper around LigBuilder is unclear. Is the grow-link approach mentioned in the methods already a component of LigBuilder, or is it custom? If it is custom, what does it do? Is the API for custom optimization routines new with the Synthesizer, or is this a component of LigBuilder? Is the genetic algorithm novel or already an existing software implementation? Is the cavity detection tool a component of LigBuilder or novel in some way? Is the fragment library utilized in the Synthesizer the default fragment library in LigBuilder, or has it been customized? Are there rules that dictate how molecule growth can occur? The scientific contribution of the Synthesizer is unclear. If there has not been any new methodological development, then it may be more appropriate to just refer to this part of the algorithm as an application layer for LigBuilder.

We appreciate Reviewer #1's constructive suggestion. We wish to emphasize that

(1) The LigBuilder software comprises various modules designed for distinct functions. The Synthesizer in Gcoupler strategically utilizes two of these modules: "CAVITY" for binding site detection and "BUILD" for de novo ligand design.

(2) While both modules are integral to LigBuilder, the Synthesizer plays a crucial role in enabling their targeted, automated, and context-aware application for GPCR drug discovery.

(3) The CAVITY module is a structure-based protein binding site detection program, which the Synthesizer employs for identifying ligand binding sites on the protein surface.

(4) The Synthesizer also leverages the BUILD module for constructing molecules tailored to the target protein, implementing a fragment-based design strategy using its integrated fragment library.

(5) The GROW and LINK methods represent two independent approaches encompassed within the aforementioned BUILD module.

**Author response image 1. sa3fig1:** Schematic representation of the key strategy used in the Synthesizer module of Gcoupler.

Our manuscript details the "grow-link" hybrid approach, which was implemented using a genetic algorithm through the following stages:

(1) Initial population generation based on a seed structure via the GROW method.

(2) Selection of "parent" molecules from the current population for inclusion in the mating pool using the LINK method.

(3) Transfer of "elite" molecules from the current population to the new population.

(4) Population expansion through structural manipulations (mutation, deletion, and crossover) applied to molecules within the mating pool.

Please note, the outcome of this process is not fixed, as it is highly dependent on the target cavity topology and the constraint parameters employed for population evaluation. Synthesizer customizes generational cycles and optimization parameters based on cavity-specific constraints, with the objective of either generating a specified number of compounds or comprehensively exploring chemical diversity against a given cavity topology.

While these components are integral to LigBuilder, Synthesizer's innovation lies

(1) in its programmatic integration and dynamic adjustment of these modules.

(2) Synthesizer distinguishes itself not by reinventing these algorithms, but by their automated coordination, fine-tuning, and integration within a cavity-specific framework.

(3) It dynamically modifies generation parameters according to cavity topology and druggability constraints, a capability not inherently supported by LigBuilder.

(4) This renders Synthesizer particularly valuable in practical scenarios where manual optimization is either inefficient or impractical.

In summary, Synthesizer offers researchers a streamlined interface, abstracting the technical complexities of LigBuilder and thereby enabling more accessible and reproducible ligand generation pipelines, especially for individuals with limited experience in structural or cheminformatics tools.

Query: (2) The use of AutoDock Vina binding energy scores to classify ligands into HABs and LABs is problematic. AutoDock Vina's energy function is primarily tuned for pose prediction and displays highly system-dependent affinity ranking capabilities. Moreover, the HAB/LAB thresholds of -7 kcal/mol or -8 kcal/mol lack justification. Were these arbitrarily selected cutoffs, or was benchmarking performed to identify appropriate cutoffs? It seems like these thresholds should be determined by calibrating the docking scores with experimental binding data (e.g., known binders with measured affinities) or through re-scoring molecules with a rigorous alchemical free energy approach.

We again express our gratitude to Reviewer #1 for these inquiries. We sincerely apologize for the lack of sufficient detail in the original version of the manuscript. In the revised manuscript, we have ensured the inclusion of a detailed rationale for every threshold utilized to prioritize high-affinity binders. Please refer to the comprehensive explanation below, as well as the revised manuscript, for further details.

We would like to clarify that:

(1) The Authenticator module is not solely reliant on absolute binding energy values for classification. Instead, it calculates binding energies for all generated compounds and applies a statistical decision-making layer to define HAB and LAB classes.

(2) Rather than using fixed thresholds, the module employs distribution-based methods, such as the Empirical Cumulative Distribution Function (ECDF), to assess the overall energy landscape of the compound set. We then applied multiple statistical tests to evaluate the HAB and LAB distributions and determine an optimal, data-specific cutoff that balances class sizes and minimizes overlap.

(3) This adaptive approach avoids rigid thresholds and instead ensures context-sensitive classification, with safeguards in place to maintain adequate representation of both classes for downstream model training, and in this way, the framework prioritizes robust statistical reasoning over arbitrary energy cutoffs and aims to reduce the risks associated with direct reliance on Vina scores alone.

(4) To assess the necessity and effectiveness of the Authenticator module, we conducted a benchmarking analysis where we deliberately omitted the HAB and LAB class labels, treating the compound pool as a heterogeneous, unlabeled dataset. We then performed random train-test splits using the Synthesizer-generated compounds and trained independent models.

(5) The results from this approach demonstrated notably poorer model performance, indicating that arbitrary or unstructured data partitioning does not effectively capture the underlying affinity patterns. These experiments highlight the importance of using the statistical framework within the Authenticator module to establish meaningful, data-driven thresholds for distinguishing High- and Low-Affinity Binders. The cutoff values are thus not arbitrary but emerge from a systematic benchmarking and validation process tailored to each dataset.

Please note: While calibrating docking scores with experimental binding affinities or using rigorous methods like alchemical free energy calculations can improve precision, these approaches are often computationally intensive and reliant on the availability of high-quality experimental data, a major limitation in many real-world screening scenarios.

In summary, the primary goal of Gcoupler is to enable fast, scalable, and broadly accessible screening, particularly for cases where experimental data is sparse or unavailable. Incorporating such resource-heavy methods would not only significantly increase computational overhead but also undermine the framework’s intended usability and efficiency for large-scale applications. Instead, our workflow relies on statistically robust, data-driven classification methods that balance speed, generalizability, and practical feasibility.

Query: (3) Neither the Results nor Methods sections provide information on how the GNNs were trained in this study. Details such as node features, edge attributes, standardization, pooling, activation functions, layers, dropout, etc., should all be described in detail. The training protocol should also be described, including loss functions, independent monitoring and early stopping criteria, learning rate adjustments, etc.

We again thank Reviewer #1 for this suggestion. We would like to mention that in the revised manuscript, we have added all the requested details. Please refer to the points below for more information.

(1) The Generator module of Gcoupler is designed as a flexible and automated framework that leverages multiple Graph Neural Network architectures, including Graph Convolutional Model (GCM), Graph Convolutional Network (GCN), Attentive FP, and Graph Attention Network (GAT), to build classification models based on the synthetic ligand datasets produced earlier in the pipeline.

(2) By default, Generator tests all four models using standard hyperparameters provided by the DeepChem framework (https://deepchem.io/), offering a baseline performance comparison across architectures. This includes pre-defined choices for node features, edge attributes, message-passing layers, pooling strategies, activation functions, and dropout values, ensuring reproducibility and consistency. All models are trained with binary cross-entropy loss and support default settings for early stopping, learning rate, and batch standardization where applicable.

(3) In addition, Generator supports model refinement through hyperparameter tuning and k-fold cross-validation (default: 3 folds). Users can either customize the hyperparameter grid or rely on Generator’s recommended parameter ranges to optimize model performance. This allows for robust model selection and stability assessment of tuned parameters.

(4) Finally, the trained models can be used to predict binding probabilities for user-supplied compounds, making it a comprehensive and user-adaptive tool for ligand screening.

Based on the reviewer #1 suggestion, we have now added a detailed description about the Generator module of Gcoupler, and also provided relevant citations regarding the DeepChem workflow.

Query: (4) GNN model training seems to occur on at most 500 molecules per training run? This is unclear from the manuscript. That is a very small number of training samples if true. Please clarify. How was upsampling performed? What were the HAB/LAB class distributions? In addition, it seems as though only synthetically generated molecules are used for training, and the task is to discriminate synthetic molecules based on their docking scores. Synthetic ligands generated by LigBuilder may occupy distinct chemical space, making classification trivial, particularly in the setting of a random split k-folds validation approach. In the absence of a leave-class-out validation, it is unclear if the model learns generalizable features or exploits clear chemical differences. Historically, it was inappropriate to evaluate ligand-based QSAR models on synthetic decoys such as the DUD-E sets - synthetic ligands can be much more easily distinguished by heavily parameterized ligand-based machine learning models than by physically constrained single-point docking score functions.

We thank reviewer #1 for these detailed technical queries. We would like to clarify that:

(1) The recommended minimum for the training set is 500 molecules, but users can add as many synthesized compounds as needed to thoroughly explore the chemical space related to the target cavity.

(2) Our systematic evaluation demonstrated that expanding the training set size consistently enhanced model performance, especially when compared to AutoDock docking scores. This observation underscores the framework's scalability and its ability to improve predictive accuracy with more training compounds.

(3) The Authenticator module initially categorizes all synthesized molecules into HAB and LAB classes. These labeled molecules are then utilized for training the Generator module. To tackle class imbalance, the class with fewer data points undergoes upsampling. This process aims to achieve an approximate 1:1 ratio between the two classes, thereby ensuring balanced learning during GNN model training.

(4) The Authenticator module's affinity scores are the primary determinant of the HAB/LAB class distribution, with a higher cutoff for HABs ensuring statistically significant class separation. This distribution is also indirectly shaped by the target cavity's topology and druggability, as the Synthesizer tends to produce more potent candidates for cavities with favorable binding characteristics.

(5) While it's true that synthetic ligands may occupy distinct chemical space, our benchmarking exploration for different sites on the same receptor still showed inter-cavity specificity along with intra-cavity diversity of the synthesized molecules.

(6) The utility of random k-fold validation shouldn't be dismissed outright; it provides a reasonable estimate of performance under practical settings where class boundaries are often unknown. Nonetheless, we agree that complementary validation strategies like leave-class-out could further strengthen the robustness assessment.

(7) We agree that using synthetic decoys like those from the DUD-E dataset can introduce bias in ligand-based QSAR model evaluations if not handled carefully. In our workflow, the inclusion of DUD-E compounds is entirely optional and only considered as a fallback, specifically in scenarios where the number of low-affinity binders (LABs) synthesized by the Synthesizer module is insufficient to proceed with model training.

(8) The primary approach relies on classifying generated compounds based on their derived affinity scores via the Authenticator module. However, in rare cases where this results in a heavily imbalanced dataset, DUD-E compounds are introduced not as part of the core benchmarking, but solely to maintain minimal class balance for initial model training. Even then, care is taken to interpret results with this limitation in mind. Ultimately, our framework is designed to prioritize data-driven generation of both HABs and LABs, minimizing reliance on synthetic decoys wherever possible.

**Author response image 2. sa3fig2:** Scatter plots depicting the segregation of High/Low-Affinity Metabolites (HAM/LAM) (indicated in green and red) identified using Gcoupler workflow with 100% training data. Notably, models trained on lesser training data size (25%, 50%, and 75% of HAB/LAB) severely failed to segregate HAM and LAM (along Y-axis). X-axis represents the binding affinity calculated using IC4-specific docking using AutoDock.

Based on the reviewer #1’s suggestion, we have now added all these technical details in the revised version of the manuscript.

Query: (5) Training QSAR models on docking scores to accelerate virtual screening is not in itself novel (see here for a nice recent example: https://www.nature.com/articles/s43588-025-00777-x), but can be highly useful to focus structure-based analysis on the most promising areas of ligand chemical space; however, we are perplexed by the motivation here. If only a few hundred or a few thousand molecules are being sampled, why not just use AutoDock Vina? The models are trained to try to discriminate molecules by AutoDock Vina score rather than experimental affinity, so it seems like we would ideally just run Vina? Perhaps we are misunderstanding the scale of the screening that was done here. Please clarify the manuscript methods to help justify the approach.

We acknowledge the effectiveness of training QSAR models on docking scores for prioritizing chemical space, as demonstrated by the referenced study (https://www.nature.com/articles/s43588-025-00777-x) on machine-learning-guided docking screen frameworks.

We would like to mention that:

(1) While such protocols often rely on extensive pre-docked datasets across numerous protein targets or utilize a highly skewed input distribution, training on as little as 1-10% of ligand-protein complexes and testing on the remainder in iterative cycles.

(2) While powerful for ultra-large libraries, this approach can introduce bias towards the limited training set and incur significant overhead in data curation, pre-computation, and infrastructure.

(3) In contrast, Gcoupler prioritizes flexibility and accessibility, especially when experimental data is scarce and large pre-docked libraries are unavailable. Instead of depending on fixed docking scores from external pipelines, Gcoupler integrates target-specific cavity detection, de novo compound generation, and model training into a self-contained, end-to-end framework. Its QSAR models are trained directly on contextually relevant compounds synthesized for a given binding site, employing a statistical classification strategy that avoids arbitrary thresholds or precomputed biases.

(4) Furthermore, Gcoupler is open-source, lightweight, and user-friendly, making it easily deployable without the need for extensive infrastructure or prior docking expertise. While not a complete replacement for full-scale docking in all use cases, Gcoupler aims to provide a streamlined and interpretable screening framework that supports both focused chemical design and broader chemical space exploration, without the computational burden associated with deep learning docking workflows.

(5) Practically, even with computational resources, manually running AutoDock Vina on millions of compounds presents challenges such as format conversion, binding site annotation, grid parameter tuning, and execution logistics, all typically requiring advanced structural bioinformatics expertise.

(6) Gcoupler's Authenticator module, however, streamlines this process. Users only need to input a list of SMILES and a receptor PDB structure, and the module automatically handles compound preparation, cavity mapping, parameter optimization, and high-throughput scoring. This automation reduces time and effort while democratizing access to structure-based screening workflows for users without specialized expertise.

Ultimately, Gcoupler's motivation is to make large-scale, structure-informed virtual screening both efficient and accessible. The model serves as a surrogate to filter and prioritize compounds before deeper docking or experimental validation, thereby accelerating targeted drug discovery.

Query: (6) The brevity of the MD simulations raises some concerns that the results may be over-interpreted. RMSD plots do not reliably compare the affinity behavior in this context because of the short timescales coupled with the dramatic topological differences between the ligands being compared; CoQ6 is long and highly flexible compared to ZST and LST. Convergence metrics, such as block averaging and time-dependent MM/GBSA energies, should be included over much longer timescales. For CoQ6, the authors may need to run multiple simulations of several microseconds, identify the longest-lived metastable states of CoQ6, and perform MM/GBSA energies for each state weighted by each state's probability.

We appreciate Reviewer #1's suggestion regarding simulation length, as it is indeed crucial for interpreting molecular dynamics (MD) outcomes. We would like to mention that:

(1) Our simulation strategy varied based on the analysis objective, ranging from short (~5 ns) runs for preliminary or receptor-only evaluations to intermediate (~100 ns) and extended (~550 ns) runs for receptor-ligand complex validation and stability assessment.

(2) Specifically, we conducted three independent 100 ns MD simulations for each receptor-metabolite complex in distinct cavities of interest. This allowed us to assess the reproducibility and persistence of binding interactions. To further support these observations, a longer 550 ns simulation was performed for the IC4 cavity, which reinforced the 100 ns findings by demonstrating sustained interaction stability over extended timescales.

(3) While we acknowledge that even longer simulations (e.g., in the microsecond range) could provide deeper insights into metastable state transitions, especially for highly flexible molecules like CoQ6, our current design balances computational feasibility with the goal of screening multiple cavities and ligands.

(4) In our current workflow, MM/GBSA binding free energies were calculated by extracting 1000 representative snapshots from the final 10 ns of each MD trajectory. These configurations were used to compute time-averaged binding energies, incorporating contributions from van der Waals, electrostatic, polar, and non-polar solvation terms. This approach offers a more reliable estimate of ligand binding affinity compared to single-point molecular docking, as it accounts for conformational flexibility and dynamic interactions within the binding cavity.

(5) Although we did not explicitly perform state-specific MM/GBSA calculations weighted by metastable state probabilities, our use of ensemble-averaged energy estimates from a thermally equilibrated segment of the trajectory captures many of the same benefits. We acknowledge, however, that a more rigorous decomposition based on metastable state analysis could offer finer resolution of binding behavior, particularly for highly flexible ligands like CoQ6, and we consider this a valuable direction for future refinement of the framework.

**Reviewer #2 (Public review):**
Summary:Query: Mohanty et al. present a new deep learning method to identify intracellular allosteric modulators of GPCRs. This is an interesting field for e.g. the design of novel small molecule inhibitors of GPCR signalling. A key limitation, as mentioned by the authors, is the limited availability of data. The method presented, Gcoupler, aims to overcome these limitations, as shown by experimental validation of sterols in the inhibition of Ste2p, which has been shown to be relevant molecules in human and rat cardiac hypertrophy models. They have made their code available for download and installation, which can easily be followed to set up software on a local machine.Strengths:Clear GitHub repositoryExtensive data on yeast systems

We sincerely thank Reviewer #2 for their thorough review, summary, and appreciation of our work. We highly value their comments and suggestions.

Weaknesses:Query: No assay to directly determine the affinity of the compounds to the protein of interest.

We thank Reviewer #2 for raising these insightful questions. During the experimental design phase, we carefully accounted for validating the impact of metabolites in the rescue response by pheromone.

We would like to mention that we performed an array of methods to validate our hypothesis and observed similar rescue effects. These assays include:

a. Cell viability assay (FDA/PI Flourometry-based)

b. Cell growth assay

c. FUN1-based microscopy assessment

d. Shmoo formation assays

e. Mating assays

f. Site-directed mutagenesis-based loss of function

g. ransgenic reporter-based assay

h. MAPK signaling assessment using Western blot.

i. And via computational techniques.

Concerning the in vitro interaction studies of Ste2p and metabolites, we made significant efforts to purify Ste2p by incorporating a His tag at the N-terminal. Despite dedicated attempts over the past year, we were unsuccessful in purifying the protein, primarily due to our limited expertise in protein purification for this specific system. As a result, we opted for genetic-based interventions (e.g., point mutants), which provide a more physiological and comprehensive approach to demonstrating the interaction between Ste2p and the metabolites.

**Author response image 3. sa3fig3:** (**a**) Affinity purification of Ste2p from *Saccharomyces cerevisiae*. Western blot analysis using anti-His antibody showing the distribution of Ste2p in various fractions during the affinity purification process. The fractions include pellet, supernatant, wash buffer, and sequential elution fractions (1–4). Wild-type and ste2Δ strains served as positive and negative controls, respectively. (**b**) Optimization of Ste2p extraction protocol. Ponceau staining (left) and Western blot analysis using anti-His antibody (right) showing Ste2p extraction efficiency. The conditions tested include lysis buffers containing different concentrations of CHAPS detergent (0.5%, 1%) and glycerol (10%, 20%).

Furthermore, in addition to the clarification above, we have added the following statement in the discussion section to tone down our claims: “A critical limitation of our study is the absence of direct binding assays to validate the interaction between the metabolites and Ste2p. While our results from genetic interventions, molecular dynamics simulations, and docking studies strongly suggest that the metabolites interact with the Ste2p-Gpa1 interface, these findings remain indirect. Direct binding confirmation through techniques such as surface plasmon resonance, isothermal titration calorimetry, or co-crystallization would provide definitive evidence of this interaction. Addressing this limitation in future work would significantly strengthen our conclusions and provide deeper insights into the precise molecular mechanisms underlying the observed phenotypic effects.”

We request Reviewer #2 to kindly refer to the assays conducted on the point mutants created in this study, as these experiments offer robust evidence supporting our claims.

Query: In conclusion, the authors present an interesting new method to identify allosteric inhibitors of GPCRs, which can easily be employed by research labs. Whilst their efforts to characterize the compounds in yeast cells, in order to confirm their findings, it would be beneficial if the authors show their compounds are active in a simple binding assay.

We express our gratitude and sincere appreciation for the time and effort dedicated by Reviewer #2 in reviewing our manuscript. We are confident that our clarifications address the reviewer's concerns.

**Reviewer #3 (Public review):**
Summary:Query: In this paper, the authors introduce the Gcoupler software, an open-source deep learning-based platform for structure-guided discovery of ligands targeting GPCR interfaces. Overall, this manuscript represents a field-advancing contribution at the intersection of AI-based ligand discovery and GPCR signaling regulation.Strengths:The paper presents a comprehensive and well-structured workflow combining cavity identification, de novo ligand generation, statistical validation, and graph neural network-based classification. Notably, the authors use Gcoupler to identify endogenous intracellular sterols as allosteric modulators of the GPCR-Gα interface in yeast, with experimental validations extending to mammalian systems. The ability to systematically explore intracellular metabolite modulation of GPCR signaling represents a novel and impactful contribution. This study significantly advances the field of GPCR biology and computational ligand discovery.

We thank and appreciate Reviewer #3 for vesting time and efforts in reviewing our manuscript and for appreciating our efforts.

**Recommendations for the authors:**

**Reviewing Editor Comments:**
We encourage the authors to address the points raised during revision to elevate the assessment from "incomplete" to "solid" or ideally "convincing." In particular, we ask the authors to improve the justification for their methodological choices and to provide greater detail and clarity regarding each computational layer of the pipeline.

We are grateful for the editors' suggestions. We have incorporated significant revisions into the manuscript, providing comprehensive technical details to prevent any misunderstandings. Furthermore, we meticulously explained every aspect of the computational workflow.

**Reviewer #2 (Recommendations for the authors):**
Query: Would it be possible to make the package itself pip installable?

Yes, it already exists under the testpip repository and we have now migrated it to the main pip. Please access the link from here: https://pypi.org/project/gcoupler/

Query: I am confused by the binding free energies reported in Supplementary Figure 8. Is the total DG reported that of the protein-ligand complex? If that is the case, the affinities of the ligands would be extremely high. They are also very far off from the reported -7 kcal/mol active/inactive cut-off.

We thank Reviewer #2 for this query. We would like to mention that we have provided a detailed explanation in the point-by-point response to Reviewer #2's original comment. Briefly, to clarify, the -7 kcal/mol active/inactive cutoff mentioned in the manuscript refers specifically to the docking-based binding free energies (ΔG) calculated using AutoDock or AutoDock Vina, which are used for compound classification or validation against the Gcoupler framework.

In contrast, the binding free energies reported in Supplementary Figure 8 are obtained through the MM-GBSA method, which provides a more detailed and physics-based estimate of binding affinity by incorporating solvation and enthalpic contributions. It is well-documented in the literature that MM-GBSA tends to systematically underestimate absolute binding free energies when compared to experimental values (10.2174/1568026616666161117112604; Table 1).

**Author response image 4. sa3fig4:** Scatter plot comparing the predicted binding affinity calculated by Docking and MM/GBSA methods, against experimental ΔG (10.1007/s10822-023-00499-0).

Our use of MM-GBSA is not to match experimental ΔG directly, but rather to assess relative binding preferences among ligands. Despite its limitations in predicting absolute affinities, MM-GBSA is known to perform better than docking for ranking compounds by their binding potential. In this context, an MM-GBSA energy value still reliably indicates stronger predicted binding, even if the numerical values appear extremely higher than typical experimental or docking-derived cutoffs.

Thus, the two energy values, docking-based and MM-GBSA, serve different purposes in our workflow. Docking scores are used for classification and thresholding, while MM-GBSA energies provide post hoc validation and a higher-resolution comparison of binding strength across compounds.

To corroborate their findings, can the authors include direct binding affinity assays for yeast and human Ste2p? This will help in establishing whether the observed phenotypic effects are indeed driven by binding of the metabolites.

We thank Reviewer #2 for raising these insightful questions. During the experimental design phase, we carefully accounted for validating the impact of metabolites in the rescue response by pheromone.

We would like to mention that we performed an array of methods to validate our hypothesis and observed similar rescue effects. These assays include:

a. Cell viability assay (FDA/PI Flourometry- based)

b. Cell growth assay

c. FUN1-based microscopy assessment

d. Shmoo formation assays

e. Mating assays

f. Site-directed mutagenesis-based loss of function

g. Transgenic reporter-based assay

h. MAPK signaling assessment using Western blot.

i. And via computational techniques.

Concerning the in vitro interaction studies of Ste2p and metabolites, we made significant efforts to purify Ste2p by incorporating a His tag at the N-terminal. Despite dedicated attempts over the past year, we were unsuccessful in purifying the protein, primarily due to our limited expertise in protein purification for this specific system. As a result, we opted for genetic-based interventions (e.g., point mutants), which provide a more physiological and comprehensive approach to demonstrating the interaction between Ste2p and the metabolites.

Furthermore, in addition to the clarification above, we have added the following statement in the discussion section to tone down our claims: “A critical limitation of our study is the absence of direct binding assays to validate the interaction between the metabolites and Ste2p. While our results from genetic interventions, molecular dynamics simulations, and docking studies strongly suggest that the metabolites interact with the Ste2p-Gpa1 interface, these findings remain indirect. Direct binding confirmation through techniques such as surface plasmon resonance, isothermal titration calorimetry, or co-crystallization would provide definitive evidence of this interaction. Addressing this limitation in future work would significantly strengthen our conclusions and provide deeper insights into the precise molecular mechanisms underlying the observed phenotypic effects.”

We request Reviewer #2 to kindly refer to the assays conducted on the point mutants created in this study, as these experiments offer robust evidence supporting our claims.

Did the authors perform expression assays to make sure the mutant proteins were similarly expressed to wt?

We thank reviewer #2 for this comment. We would like to mention that:

(1) In our mutants (S75A, T155D, L289K)-based assays, all mutants were generated using integration at the same chromosomal TRP1 locus under the GAL1 promoter and share the same C-terminal CYC1 terminator sequence used for the reconstituted wild-type (rtWT) construct, thus reducing the likelihood of strain-specific expression differences.

(2) Furthermore, all strains were grown under identical conditions using the same media, temperature, and shaking parameters. Each construct underwent the same GAL1 induction protocol in YPGR medium for identical durations, ensuring uniform transcriptional activation across all strains and minimizing culture-dependent variability in protein expression.

(3) Importantly, both the rtWT and two of the mutants (T155D, L289K) retained α-factor-induced cell death (PI and FUN1-based fluorometry and microscopy; Figure 4c-d) and MAPK activation (western blot; Figure 4e), demonstrating that the mutant proteins are expressed at levels sufficient to support signalling.

**Reviewer #3 (Recommendations for the authors):**
My comments that would enhance the impact of this method are:(1) While the authors have compared the accuracy and efficiency of Gcoupler to AutoDock Vina, one of the main points of Gcoupler is the neural network module. It would be beneficial to have it evaluated against other available deep learning ligand generative modules, such as the following: 10.1186/s13321-024-00829-w, 10.1039/D1SC04444C.

Thank you for the observation. To clarify, our benchmarking of Gcoupler’s accuracy and efficiency was performed against AutoDock, not AutoDock Vina. This choice was intentional, as AutoDock is one of the most widely used classical techniques in computer-aided drug design (CADD) for obtaining high-resolution predictions of ligand binding energy, binding poses, and detailed atomic-level interactions with receptor residues. In contrast, AutoDock Vina is primarily optimized for large-scale virtual screening, offering faster results but typically with lower resolution and limited configurational detail.

Since Gcoupler is designed to balance accuracy with computational efficiency in structure-based screening, AutoDock served as a more appropriate reference point for evaluating its predictions.

We agree that benchmarking against other deep learning-based ligand generative tools is important for contextualizing Gcoupler’s capabilities. However, it's worth noting that only a few existing methods focus specifically on cavity- or pocket-driven de novo drug design using generative AI, and among them, most are either partially closed-source or limited in functionality.

While PocketCrafter (10.1186/s13321-024-00829-w) offers a structure-based generative framework, it differs from Gcoupler in several key respects. PocketCrafter requires proprietary preprocessing tools, such as the MOE QuickPrep module, to prepare protein pocket structures, limiting its accessibility and reproducibility. In addition, PocketCrafter’s pipeline stops at the generation of cavity-linked compounds and does not support any further learning from the generated data.

Similarly, DeepLigBuilder (10.1039/D1SC04444C) provides de novo ligand generation using deep learning, but the source code is not publicly available, preventing direct benchmarking or customization. Like PocketCrafter, it also lacks integrated learning modules, which limits its utility for screening large, user-defined libraries or compounds of interest.

Additionally, tools like AutoDesigner from Schrödinger, while powerful, are not publicly accessible and hence fall outside the scope of open benchmarking.

**Author response table 1. sa3table1:** Comparison of de novo drug design tools. SBDD refers to Structure-Based Drug Design, and LBDD refers to Ligand-Based Drug Design.

Tools	Drug design Type	Open Source	Code Availability	End-to-End Solution	Command Line support
Gcoupler	SBDD+LBDD	Yes	Yes	Yes	Yes
Pocket Crafter	SBDD	Partially	Partially	No	Yes
DeepLigBuilder	SBDD	--	No	--	--
AutoDesigner	LBDD	No	No	--	Yes

In contrast, Gcoupler is a fully open-source, end-to-end platform that integrates both Ligand-Based and Structure-Based Drug Design. It spans from cavity detection and molecule generation to automated model training using GNNs, allowing users to evaluate and prioritize candidate ligands across large chemical spaces without the need for commercial software or advanced coding expertise.

(2) In Figure 2, the authors mention that IC4 and IC5 potential binding sites are on the direct G protein coupling interface ("This led to the identification of 17 potential surface cavities on Ste2p, with two intracellular regions, IC4 and IC5, accounting for over 95% of the Ste2p-Gpa1p interface (Figure 2a-b, Supplementary Figure 4j-n)..."). Later, however, in Figure 4, when discussing which residues affect the binding of the metabolites the most, the authors didn't perform MD simulations of mutant STE2 and just Gpa1p (without metabolites present). It would be beneficial to compare the binding of G protein with and without metabolites present, as these interface mutations might be affecting the binding of G protein by itself.

Thank you for this insightful suggestion. While we did not perform in silico MD simulations of the mutant Ste2-Gpa1 complex in the absence of metabolites, we conducted experimental validation to functionally assess the impact of interface mutations. Specifically, we generated site-directed mutants (S75A, L289K, T155D) and expressed them in a ste2Δ background to isolate their effects.

As shown in the Supplementary Figure, these mutants failed to rescue cells from α-factor-induced programmed cell death (PCD) upon metabolite pre-treatment. This was confirmed through fluorometry-based viability assays, FUN1 staining, and p-Fus3 signaling analysis, which collectively monitor MAPK pathway activation (Figure 4c–e).

Importantly, the induction of PCD in response to α-factor in these mutants demonstrates that G protein coupling is still functionally intact, indicating that the mutations do not interfere with Gpa1 binding itself. However, the absence of rescue by metabolites strongly suggests that the mutated residues play a direct role in metabolite binding at the Ste2p–Gpa1p interface, thus modulating downstream signaling.

While further MD simulations could provide structural insight into the isolated mutant receptor–G protein interaction, our experimental data supports the functional relevance of metabolite binding at the identified interface.

(3) While the experiments, performed by the authors, do support the hypothesis that metabolites regulate GPCR signaling, there are no experiments evaluating direct biophysical measurements (e.g., dissociation constants are measured only in silico).

We thank Reviewer #3 for raising these insightful comments. We would like to mention that we performed an array of methods to validate our hypothesis and observed similar rescue effects. These assays include:

a. Cell viability assay (FDA/PI Flourometry- based)

b. Cell growth assay

c. FUN1-based microscopy assessment

d. Shmoo formation assays

e. Mating assays

f. Site-directed mutagenesis-based loss of function

g. Transgenic reporter-based assay

h. MAPK signaling assessment using Western blot.

i. And via computational techniques.

Concerning the direct biophysical measurements of Ste2p and metabolites, we made significant efforts to purify Ste2p by incorporating a His tag at the N-terminal, with the goal of performing Microscale Thermophoresis (MST) and Isothermal Titration Calorimetry (ITC) measurements. Despite dedicated attempts over the past year, we were unsuccessful in purifying the protein, primarily due to our limited expertise in protein purification for this specific system. As a result, we opted for genetic-based interventions (e.g., point mutants), which provide a more physiological and comprehensive approach to demonstrating the interaction between Ste2p and the metabolites.

Furthermore, in addition to the clarification above, we have added the following statement in the discussion section to tone down our claims: “A critical limitation of our study is the absence of direct binding assays to validate the interaction between the metabolites and Ste2p. While our results from genetic interventions, molecular dynamics simulations, and docking studies strongly suggest that the metabolites interact with the Ste2p-Gpa1 interface, these findings remain indirect. Direct binding confirmation through techniques such as surface plasmon resonance, isothermal titration calorimetry, or co-crystallization would provide definitive evidence of this interaction. Addressing this limitation in future work would significantly strengthen our conclusions and provide deeper insights into the precise molecular mechanisms underlying the observed phenotypic effects.”

(4) The authors do not discuss the effects of the metabolites at their physiological concentrations. Overall, this manuscript represents a field-advancing contribution at the intersection of AI-based ligand discovery and GPCR signaling regulation.

We thank reviewer #3 for this comment and for recognising the value of our work. Although direct quantification of intracellular free metabolite levels is challenging, several lines of evidence support the physiological relevance of our test concentrations.

- Genetic validation supports endogenous relevance: Our genetic screen of 53 metabolic knockout mutants showed that deletions in biosynthetic pathways for these metabolites consistently disrupted the α-factor-induced cell death, with the vast majority of strains (94.4%) resisting the α-factor-induced cell death, and notably, a subset even displayed accelerated growth in the presence of α‑factor. This suggests that endogenous levels of these metabolites normally provide some degree of protection, supporting their physiological role in GPCR regulation.

- Metabolomics confirms in vivo accumulation: Our untargeted metabolomics analysis revealed that α-factor-treated survivors consistently showed enrichment of CoQ6 and zymosterol compared to sensitive cells. This demonstrates that these metabolites naturally accumulate to protective levels during stress responses, validating their biological relevance.